# Understanding how junction resistances impact the conduction mechanism in nano-networks

Cian Gabbett [1,9], Adam G. Kelly[1,2,9], Emmet Coleman [1], Luke Doolan[1], Tian Carey [1], Kevin Synnatschke[1], Shixin Liu [1], Anthony Dawson[1], Domhnall O'Suilleabhain[1], Jose Munuera [1,3], Eoin Caffrey [1], John B. Boland[1], Zdeněk Sofer [4], Goutam Ghosh[5], Sachin Kinge[6], Laurens D. A. Siebbeles [5], Neelam Yadav [7], Jagdish K. Vij [7], Muhammad Awais Aslam [8], Aleksandar Matkovic [8] & Jonathan N. Coleman [1] ✉

Networks of nanowires, nanotubes, and nanosheets are important for many applications in printed electronics. However, the network conductivity and mobility are usually limited by the resistance between the particles, often referred to as the junction resistance. Minimising the junction resistance has proven to be challenging, partly because it is difficult to measure. Here, we develop a simple model for electrical conduction in networks of 1D or 2D nanomaterials that allows us to extract junction and nanoparticle resistances from particle-size-dependent DC network resistivity data. We find junction resistances in porous networks to scale with nanoparticle resistivity and vary from 5 Ω for silver nanosheets to 24 GΩ for $WS_2$ nanosheets. Moreover, our model allows junction and nanoparticle resistances to be obtained simultaneously from AC impedance spectra of semiconducting nanosheet networks. Through our model, we use the impedance data to directly link the high mobility of aligned networks of electrochemically exfoliated $MoS_2$ nanosheets (≈ 7 cm² V⁻¹ s⁻¹) to low junction resistances of ~2.3 MΩ. Temperature-dependent impedance measurements also allow us to comprehensively investigate transport mechanisms within the network and quantitatively differentiate intra-nanosheet phonon-limited bandlike transport from inter-nanosheet hopping.

Printed electronic devices are increasingly important due to their flexibility, scalability, and cost-effectiveness[1]. Driven by their combination of solution-processability and strong electrical performance, 0D, 1D, and 2D nanoparticles have now been widely explored as materials for printed electronics[1,2]. Printed networks of carbon nanotubes have shown great promise for use in transistors[3,4], LEDs[5], and photodetectors[6], while metallic nanowire networks have been heavily studied as transparent electrodes[7,8], EMI shields[9], and heating

¹School of Physics, CRANN & AMBER Research Centres, Trinity College Dublin, Dublin 2, Ireland. ²i3N/CENIMAT, Faculty of Science and Technology, Universidade NOVA de Lisboa, Campus de Caparica, 2829-516 Caparica, Portugal. ³Department of Physics, Faculty of Sciences, University of Oviedo, C/ Leopoldo Calvo Sotelo, 18, 33007 Oviedo, Asturias, Spain. ⁴Department of Inorganic Chemistry, University of Chemistry and Technology Prague, Technická 5, Prague 6 166 28, Czech Republic. ⁵Chemical Engineering Department, Delft University of Technology, Van der Maasweg 9, NL-2629 HZ Delft, The Netherlands. ⁶Materials Research & Development, Toyota Motor Europe, B1930 Zaventem, Belgium. ⁷Department of Electronic & Electrical Engineering, Trinity College Dublin, Dublin 2, Ireland. ⁸Chair of Physics, Department Physics, Mechanics and Electrical Engineering, Montanuniversität Leoben, Franz Josef Strasse 18, 8700 Leoben, Austria. ⁹These authors contributed equally: Cian Gabbett, Adam G. Kelly. ✉e-mail: colemaj@tcd.ie

elements[10]. More recently, solution-processed networks of 2D materials, such as graphene and $MoS_2$, have been investigated for a broad range of applications in all areas of (opto)electronics and energy storage[11–13].

The success of this approach relies on exploiting the exceptional intrinsic properties of the individual nanoparticles (e.g. their high conductivity or carrier mobility) when they are assembled into large area networks. Conductive nanoparticles have been printed into networks with conductivities reaching $\approx 10^5$ S m$^{-1}$ for graphene[12], $\approx 10^6$ S m$^{-1}$ for MXenes[14], $>10^6$ S m$^{-1}$ for silver nanowires (AgNWs)[7,15], and $>10^7$ S m$^{-1}$ for silver nanosheets (AgNSs)[16]. While the properties of conductive networks have approached those of their constituent nanoparticles, semiconducting networks have shown less progress. Semiconducting carbon nanotube networks typically display mobilities $<100$ cm$^2$ V$^{-1}$ s$^{-1}$ [17] compared to $>10^5$ cm$^2$ V$^{-1}$ s$^{-1}$ for an individual nanotube[18]. Similarly, networks of solution-processed $MoS_2$ nanosheets[12,19–21] achieve $\approx 10$ cm$^2$ V$^{-1}$ s$^{-1}$ compared to $>100$ cm$^2$ V$^{-1}$ s$^{-1}$ for mechanically exfoliated nanosheets[22–24]. The reason for these discrepancies is simple: almost all networks are believed to be limited by the junctions between particles, with semiconducting networks being particularly junction-limited[12,17].

Junction-limited networks can be classed as those where the inter-nanoparticle junction resistance, $R_J$, is greater than the intrinsic resistance of the constituent nanoparticles, $R_{NP}$, i.e. $R_J > R_{NP}$. Realising high performance printed devices requires minimising the junction resistance relative to the nanoparticle resistance. This makes the fabrication of networks with $R_J < R_{NP}$ [12,25,26], such that the network properties approach those of the individual nanoparticles, an important goal. Strategies to achieve this include optimising nanoparticle dimensions or deposition techniques[12], and chemical cross-linking[25,26]. However, without the ability to measure $R_J$ and $R_{NP}$, assessing the progress of various strategies towards achieving $R_J < R_{NP}$ is difficult.

On the other hand, due to our inability to easily measure either the junction or the nanoparticle resistance in situ, even proving that low mobility is due to junction limitations is challenging. For example, one might argue that processing can introduce defects into the nanoparticles which reduces their intrinsic mobility (although this is unlikely, as we argue in Supplementary Note 3). This would decrease the network mobility even for a negligible junction resistance. Thus, to fully understand the reason why the network mobility is lower than that of the nanoparticles, one must be able to measure $R_J$ and $R_{NP}$ to determine which is larger and pinpoint the limiting factor.

Despite their importance, the literature contains very little quantitative data on junction resistances. While conductive-AFM[27] or utilisation of micro-electrodes[28] can yield local information on both nanoparticle and junction resistances, these methods are unsuitable for large area printed networks or in-device measurements. This lack of basic information has hindered printed device development and forced a reliance on trial-and-error for device optimisation.

Another approach to finding $R_J$ involves using models to link network conductivity to junction resistance, specifically for nanowire networks[29–31]. However, we believe it would be useful to develop a set of simple analytical equations which can be applied to nanoparticles beyond 1D nanowires i.e. 2D nanosheets and even 0D nanodots. Such equations could be used to fit data for the resistivity of 2D, 1D, and 0D networks, versus parameters such as nanoparticle size, yielding values for $R_J$ and $R_{NP}$ as fit parameters. In addition, access to suitable equations could allow one to directly link the network properties to those of a single (average) nanoparticle-junction pair. As we will show, such a link allows the development of new methodologies to analyse junctions.

In this work, we develop a simple model relating the resistivity and mobility of nanoparticle networks to controllable nanoparticle parameters and network properties, including junction resistance. We show that this model accurately describes experimental data for various nanomaterials and allows the extraction of both nanoparticle and

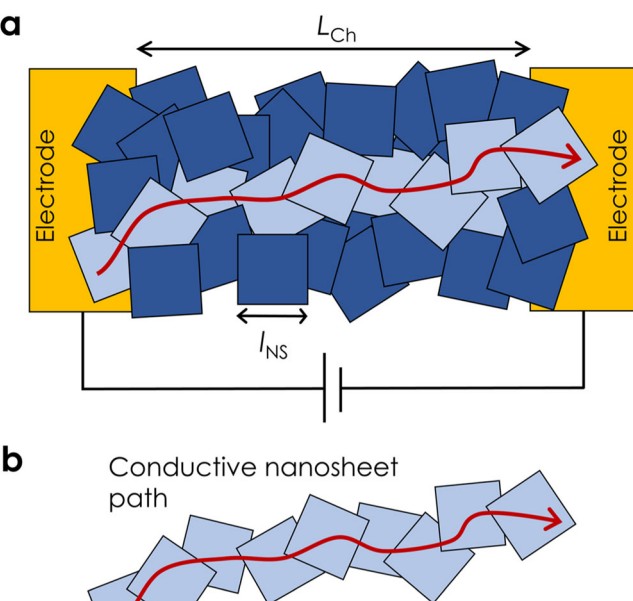

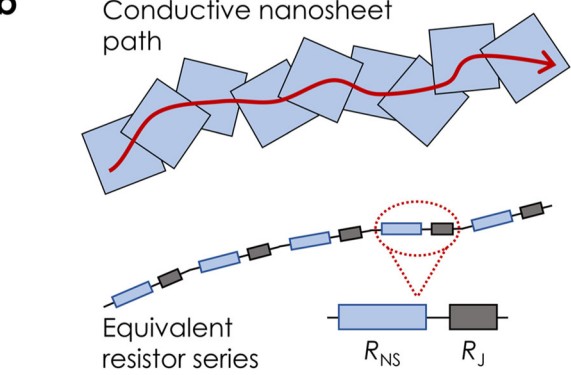

**Fig. 1 | Model schematics. a** Schematic illustrating a nanosheet network connected to two electrodes with channel length, $L_{Ch}$, under an applied voltage. A single conducting path consisting of a linear array of nanosheets is shown spanning the channel length (red arrow). The nanosheet lateral size is $l_{NS}$. While this schematic depicts a nanosheet network, a similar diagram could easily be produced to represent a nanowire network. **b** This conducting path can be considered as a chain of resistor pairs, with each pair consisting of a resistance representing a nanosheet, $R_{NS}$, and one representing the inter-sheet junction, $R_J$.

junction resistances. We combine this model with impedance spectroscopy measurements to develop a powerful technique for simultaneously measuring both nanosheet and junction resistances within networks of semiconducting nanosheets.

## Results and discussion
### Model development
We utilise a circuit-based approach to derive an equation for the resistivity of networks of 2D, 1D, or 0D nanoparticles (e.g., nanosheets, nanowires, or nanodots), $\rho_{Net}$, in terms of the properties of individual nanoparticles, as well as the junction resistance, $R_J$, and network porosity, $P_{Net}$ (see Supplementary Note 1 for full derivation). We consider the network as consisting of many well-defined conductive paths in parallel. Within a given current path (Fig. 1a), we assume each carrier passes through a linear array of nanoparticles, during which it must cross an inter-particle junction every time it traverses a nanoparticle (Fig. 1b). Thus, an individual current path can then be modelled as a linear array of nanoparticle-junction pairs, with each pair described by two resistors representing the average nanoparticle ($R_{NP}$) and junction ($R_J$) resistances (Fig. 1b).

By relating the number of resistor pairs in a path to the channel length, $L_{Ch}$, and the average distance travelled within each nanoparticle, one can estimate the typical potential drops across individual nanoparticles and junctions upon application of a voltage. These potential drops yield the average transit times across individual nanoparticles and junctions (Supplementary Note 1). Combining these equations with an expression for the total transit time through the

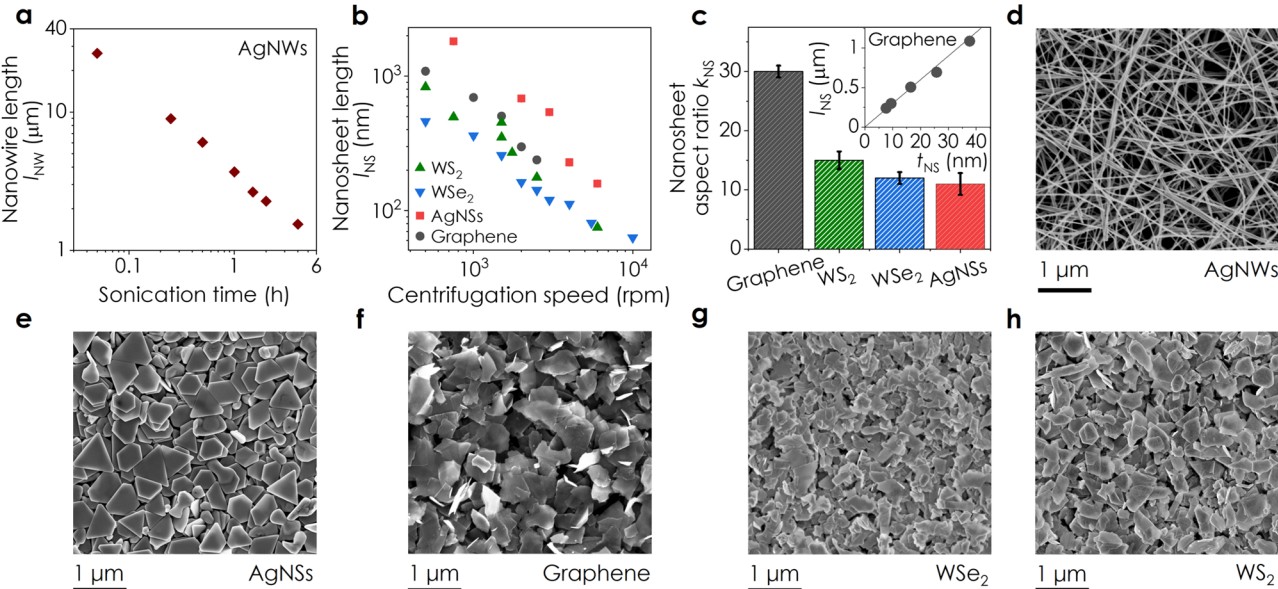

**Fig. 2 | Microscopic analysis of nanoparticles and networks. a** Mean length of silver nanowires (AgNWs) size-selected by sonication-induced scission as a function of sonication time. The uncertainty in $l_{NW}$ is ±standard error (SE) in the mean ($n = 100–200$). **b** Mean nanosheet length, $l_{NS}$, of centrifuge-fractionated silver nanosheets (AgNSs), graphene, WS₂ and WSe₂ nanosheets, plotted versus centrifugation speed. The uncertainty in $l_{NS}$ is ±SE in the mean for the AgNSs ($n = 135–251$), graphene ($n = 210–270$), WS₂ ($n = 226–443$) and WSe₂ ($n = 89–227$). **c** Average nanosheet aspect ratio, $k_{NS}$, across all size-selected fractions for each 2D

material. Inset: Exemplary data showing linear scaling between nanosheet length and thickness, $t_{NS}$, over five size-selected fractions of LPE graphene, consistent with an aspect ratio of $k_{NS} \approx 30$ (See Supplementary Note 2 for all data). The uncertainty in $k_{NS}$ is ±the root sum of squares (RSS) of SE in the mean for $l_{NS}$ and $t_{NS}$ ($n = 89–443$). **d** Surface SEM image of a spray-cast network of AgNWs. Representative surface SEM images of spray-cast networks of (**e**) AgNSs, (**f**) graphene, (**g**) WSe₂ and (**h**) WS₂ nanosheets.

channel, one can obtain an equation for the network mobility, $\mu_{Net}$:

$$\mu_{Net} \approx \frac{\mu_{NP}}{\left[1 + \frac{R_J}{R_{NP}}\right]\left[1 + \frac{2}{n_{NP}l_{NP}A_{NP}}\right]} \quad (1)$$

where $\mu_{NP}$, $n_{NP}$, $l_{NP}$, and $A_{NP}$ are the nanoparticle mobility, carrier density, length, and cross-sectional area. This equation clearly shows that $\mu_{Net}$ depends on $R_J/R_{NP}$, which should be minimised to maximise mobility. In addition, a minimal rearrangement of Eq. (1) shows $\mu_{Net}^{-1} \propto (R_J + R_{NP})$, in line with previous proposals[12,32,33].

We can generate equations for network resistivity, $\rho_{Net}$, specific to 1D nanowires/nanotubes and 2D nanosheets by combining Eq. (1) with an expression for network resistivity[12], $\rho_{Net}^{-1} = (1 - P_{Net})n_{NP}e\mu_{Net}$, where $P_{Net}$ is the network porosity (see Supplementary Note 1, Supplementary Section 1.2 for detailed derivation). In addition, we utilise dimensionality-specific equations relating $R_{NP}$ to $l_{NP}$ and $A_{NP}$. Strictly speaking, we define the nanoparticle resistance as the resistance of the portion of the nanoparticle through which current flows on average. This leads to the equations relevant to 1D ($R_{NW} = \rho_{NW}(l_{NW}/2)/(\pi D_{NW}^2/4)$) and 2D ($R_{NS} = \rho_{NS}/(2t_{NS})$) nanoparticles (see Supplementary Note 1, Supplementary Sections 1.3 and 1.4). The geometry-specific subscripts NS and NW refer to a nanosheet and nanowire, respectively. Here, $D_{NW}$ and $l_{NW}$ are the nanowire diameter and length, $t_{NS}$ is the nanosheet thickness and $\rho_{NW}$ and $\rho_{NS}$ represent the individual nanowire and nanosheet resistivities. This results in the following equations for network resistivity where Equations (2) and (3) apply to 1D and 2D particles, respectively (see Supplementary Note 1, Supplementary Section 1.5 for the 0D expression):

$$\rho_{Net} \approx \frac{1}{(1 - P_{Net})}\left[\rho_{NW} + \frac{\pi D_{NW}^2 R_J}{2l_{NW}}\right]\left[1 + \frac{8}{n_{NW}l_{NW}\pi D_{NW}^2}\right] \quad (1D) \quad (2)$$

$$\rho_{Net} \approx \frac{\left[\rho_{NS} + 2t_{NS}R_J\right]}{(1 - P_{Net})}\left[1 + \frac{2}{n_{NS}t_{NS}l_{NS}^2}\right] \quad (2D) \quad (3)$$

It is important to note that for large values of $n_{NW}$ and $n_{NS}$ such as those found for graphene, AgNSs, AgNWs, or heavily doped semiconductors, the second square-bracketed terms in Eqs. (2) and (3) approximate to 1 and can be ignored.

While no physics-based models for nanosheet network resistivity exist, we can compare Eq. (2) to a previously reported model for metallic nanowire networks[29]. In Supplementary Note 1 (Supplementary Section 1.7), we show that the equation for the network sheet resistance reported in ref. 29 can be rearranged to give an equation for $\rho_{Net}$ that has properties virtually identical to Eq. (2). This supports the validity of our approach.

## Measuring the dependence of network resistivity on nanoparticle dimensions

Equations (2) and (3) suggest a rich array of size-dependent behaviour that has not yet been observed in nanomaterial networks (Supplementary Note 1). For example, the appearance of nanosheet size parameters (i.e., $t_{NS}$ and $l_{NS}$) in both the denominator and numerator of Eq. (3) predicts a non-monotonic size-dependence with either a positive or negative $d\rho_{Net}/dt_{NS}$. To search for such behaviour and to test the validity of Eqs. (2) and (3), we produced inks of 1D AgNWs and four types of 2D nanosheets: graphene, WS₂, and WSe₂ (synthesised by liquid-phase exfoliation, LPE[34]), and commercial AgNSs (see Fig. 2, and Supplementary Note 2 for a full characterisation of each material). Each material was size-selected into fractions (Fig. 2a–c) which were then spray-coated to produce a set of networks for electrical testing, with representative SEM images shown in Fig. 2d–h. All networks were thick enough to be in the thickness-independent conductivity regime[35].

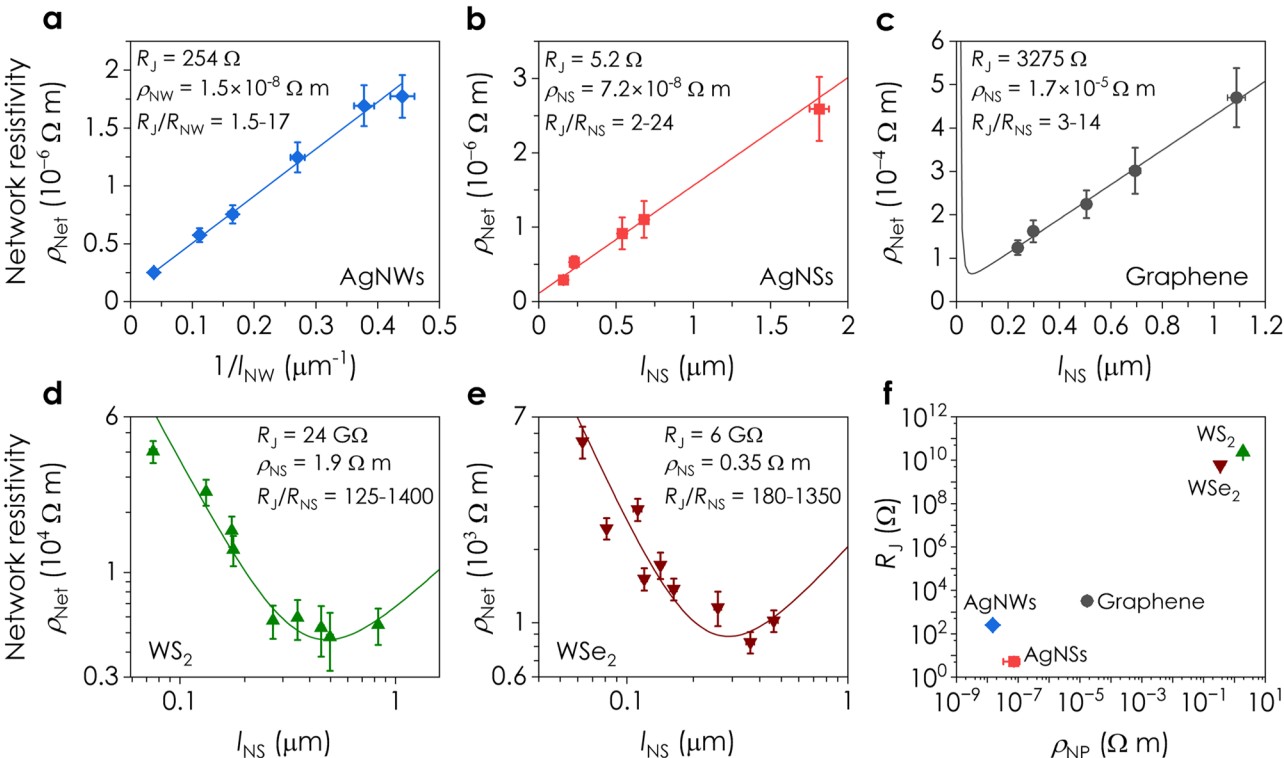

**Fig. 3 | Dependence of network resistivity on nanoparticle dimensions.**
**a** Resistivity of spray-cast silver nanowire (AgNW) networks versus inverse nanowire length, $l_{NW}^{-1}$. The line is a fit to Eq. (2). Here, the carrier density is large, allowing the second square bracketed term in Eq. (2) to be neglected. The uncertainty in $l_{NW}^{-1}$ is ±SE in the mean ($n = 100–200$) and $\rho_{Net}$ is the RSS of errors in network cross sectional area, $A_{Net}$, and $L_{Ch}$. Resistivity of spray-cast nanosheet networks, $\rho_{Net}$, versus nanosheet length, $l_{NS}$, for networks of (**b**) AgNSs, (**c**) graphene, (**d**) WS$_2$ and (**e**) WSe$_2$. In **b–e**, the lines represent fits to Eq. (3). The carrier density is large in **b** and **c** allowing the second square-bracketed term in Eq. (3) to be neglected. The behaviour in **b** and **c** is counterintuitive as the general expectation is that smaller nanosheets lead to higher resistivity. Fitting the data in **a–e** yields values for the junction resistance, $R_J$, and nanoparticle resistivity, $\rho_{NP}$, for each material. The latter parameter, combined with the nanoparticle dimensions yields the nanoparticle resistance, $R_{NP}$. Values of $R_J$ and $\rho_{NP}$ $\rho_{NP}$, as well as ranges of $R_J/R_{NP}$, are given for each material in **a–e** and Table 1. The data are presented as means ± SE in the mean for $l_{NS}$ ($n = 89–443$, Supplementary Note 2) and $\rho_{Net}$ ($n = 5–29$). **f** Junction resistance, $R_J$, plotted versus nanoparticle resistivity, $\rho_{NP}$, demonstrating scaling. The uncertainty in $R_J$ is ±the error in the fits to Eqs. (2) and (3). The uncertainty in $\rho_{NP}$ is the RSS of errors in $R_J$ and SE in the mean for $t_{NS}$ (or $D_{NW}$) across each material ($n = 135–443$).

The measured size-dependent DC resistivity is shown for all five materials in Fig. 3a–e. Because $n_{NW}$ is large for AgNWs, Eq. (2) predicts that $\rho_{Net}$ scales linearly with $l_{NW}^{-1}$, behaviour that is clearly seen in Fig. 3a. Nanosheets produced by LPE[36] and the AgNSs display a roughly constant aspect ratio, $k_{NS}$ (see the size distributions in Supplementary Note 2), allowing us to reduce the number of variables in Eq. (3) by replacing $t_{NS}$ with $t_{NS} = l_{NS}/k_{NS}$. Neglecting the final term in Eq. (3) for graphene and AgNSs, we now find that $\rho_{Net}$ should scale linearly with $l_{NS}$, as seen experimentally in Fig. 3b, c. However, for semiconducting materials, $n_{NS}$ is low meaning the final term in Eq. (3) must be considered, resulting in the prediction of a resistivity-minimum at a specific nanosheet size. Figure 3d, e show $\rho_{Net}$ for WS$_2$ and WSe$_2$, which initially falls with increasing $l_{NS}$, before reaching a minimum, behaviour that is consistent with our non-intuitive prediction. That such a minimum exists is important as it suggests the existence of an optimal nanosheet size where the network resistivity is minimised. We argue in Supplementary Note 3 that these materials show no significant variations of intrinsic nanosheet properties with size (e.g. due to the presence of sonication-induced defects) that might contribute to the observed size-dependent effects.

Equations (2) and (3) describe our data well, including the counterintuitive trends for WS$_2$ and WSe$_2$, with fitting yielding values for $R_J$ and $\rho_{NP}$ as shown in Fig. 3a–e and Table 1. For each nanoparticle size, we can convert $\rho_{NP}$ to $R_{NP}$ allowing us to also report $R_J/R_{NP}$ for each material in Fig. 3a–e. Our data yield $\rho_{NW} = (1.5 \pm 0.2) \times 10^{-8}$ Ω m for the AgNWs and $\rho_{NS} = (7.2 \pm 3.9) \times 10^{-8}$ Ω m for the AgNSs, both close to bulk silver ($1.6 \times 10^{-8}$ Ω m). For graphene, the nanosheet resistivity was

found to be $\rho_{NS} = (1.7 \pm 0.5) \times 10^{-5}$ Ω m, consistent with in-plane graphite ($\approx 10^{-5} – 10^{-6}$ Ω m)[37]. The $\rho_{NS}$ values for WS$_2$ (1.9 ± 0.3) Ω m and WSe$_2$ (0.35 ± 0.07) Ω m are consistent with previously reported values of 0.6 Ω m[38] and 0.1 Ω m[39], respectively. The values of $R_J$ ranged from 5.2 ± 0.7 Ω for AgNSs to 24 ± 2.3 GΩ for WS$_2$ and are consistent with previous estimates of $R_J \approx 3$ Ω for AgNSs[16], 185 Ω for AgNWs[40], $\approx 10^3 – 10^5$ Ω for graphene[12], and $\approx 10^9$ Ω for MoS$_2$[32]. The extracted values for each material are summarised in Table 1.

These data clearly show that the $R_J/R_{NP}$ values were >1 for each material (Fig. 3a–e), indicating that all of these networks were predominantly junction-limited. In addition, we can summarise our results for the various materials by plotting $R_J$ versus nanoparticle resistivity, $\rho_{NP}$, as shown in Fig. 3f. Interestingly, this graph shows a clear relationship between $R_J$ and $\rho_{NP}$, especially for the 2D materials. The metallic nanoparticles have very low junction resistances with $R_J$ on the order of Ohms, the semimetal (graphene) showing $R_J$ on the order of kOhms, and the semiconductors with $R_J$ on the order of GOhms. This implies a relationship between $R_J$ and nanoparticle band structure, likely via the details of the inter-particle potential barrier.

## A simultaneous measurement of $R_J$ and $R_{NP}$ using impedance spectroscopy

Measuring $R_{NP}$ and $R_J$ as described above is time-consuming because it requires extensive sample preparation in the form of the size-selection procedure. We propose that AC impedance spectroscopy, a powerful tool for device and materials characterisation[41,42], can leverage the intrinsic capacitance associated with each junction to extract

**Table 1 | Fitting parameters from the DC network resistivity and AC impedance models in this work**

| | $\mu_{NP}$ (cm² (Vs)⁻¹) | $n_{NP}$ (m⁻³) | $\rho_{NP}$ (Ω m) | $k_{NS}$ or $D_{NW}$ | $R_J$ (Ω) | $P_{Net}$ | $A_{Net}$ (m²) | $L_{Ch}$ (m) |
|---|---|---|---|---|---|---|---|---|
| LPE Graphene | **2000** | $(1.8 \pm 0.6) \times 10^{24}$ | $(1.7 \pm 0.5) \times 10^{-5}$ | **30** | $3275 \pm 314$ | **0.45** | **$1.4 \times 10^{-9}$** | **$(1.4–20) \times 10^{-3}$** |
| AgNSs | – | – | $(7.2 \pm 3.9) \times 10^{-8}$ | **11** | $5.2 \pm 0.7$ | **0.35** | **$2.2 \times 10^{-9}$** | **$3 \times 10^{-3}$** |
| LPE WS₂ | **60** | $(5.4 \pm 0.9) \times 10^{20}$ | $1.9 \pm 0.3$ | **15** | $(2.4 \pm 0.2) \times 10^{10}$ | **0.5** | **$4.9 \times 10^{-8}$** | **$5 \times 10^{-5}$** |
| LPE WSe₂ | **90** | $(2.0 \pm 0.4) \times 10^{21}$ | $0.35 \pm 0.07$ | **12** | $(6.1 \pm 0.7) \times 10^{9}$ | **0.5** | **$3.2 \times 10^{-9}$** | **$8.5 \times 10^{-5}$** |
| AgNWs | – | – | $(1.5 \pm 0.2) \times 10^{-8}$ | **39 nm** | $254 \pm 8$ | **0.85** | **$2.4 \times 10^{-10}$** | **$3.5 \times 10^{-2}$** |
| EE MoS₂ | $37 \pm 4$ | **$(3.8 \pm 0.4) \times 10^{23}$** | $(4.4 \pm 0.5) \times 10^{-3}$ | **303** | $(2.9 \pm 0.1) \times 10^{6}$ | **0.02** | **$2.9 \times 10^{-10}$** | **$5 \times 10^{-5}$** |

Fixed and measured parameters are in bold. All other values were extracted from fits to Eq. (2) (AgNWs) and Eq. (3) (AgNSs, LPE graphene, WS₂ and WSe₂) for the DC data, and from fits to Eq. (5) (EE MoS₂) for the AC data. Nanosheet aspect ratio, $k_{NS}$, and AgNW diameter, $D_{NW}$, were measured using AFM and SEM respectively. Network porosity, $P_{Net}$, values were taken from Gabbett et al.[47] and Carey et al.[19]. WS₂ and WSe₂ nanosheet mobilities were extracted from Kelly et al.[53], while the mobility of graphene was taken as the in-plane mobility of graphite[83]. The network channel length, $L_{Ch}$, and cross-sectional area, $A_{Net}$, were determined using the known electrode dimensions for each material, as well as profilometry and AFM measurements. For LPE materials, values for the nanoparticle resistivity, $\rho_{NP}$, were calculated from nanoparticle mobility, $\mu_{NP}$, and carrier density, $n_{NP}$, values using $\rho_{NP} = (n_{NP}e\mu_{NP})^{-1}$. For AgNWs and AgNSs, $n_{NP}$ is very large, allowing the second square-bracketed terms in Eqs. (2) and (3) to be neglected. Then a simple linear fit can be used yielding $\rho_{NP}$ and $R_J$, once $P_{Net}$ and $k_{NS}$ (or $D_{NW}$) are known. For the electrochemically exfoliated MoS₂, $R_{NS}$ and $R_J$ were extracted from the impedance fit. $R_{NS}$ was then converted to $\rho_{NS}$ via the nanosheet thickness. The measured carrier density was then used to convert $\rho_{NS}$ to $\mu_{NS}$.

information about nanosheet and junction resistances (Supplementary Note 4). A similar approach has been utilised for both grain/grain-boundary[43–45] and 2D systems[46]. However, because such measurements probe all junctions in all current paths, these measurements have up to now yielded $R_{NP}$ and $R_J$ in arbitrary units (but not absolute values), limiting useful analysis.

To extract absolute values for $R_{NS}$ and $R_J$ for nanosheet networks, the impedance spectra of the network ($Z_{Net}$) must be converted to spectra representing the average nanosheet-junction pair ($Z_{NS-J}$) within the network. These nanosheet-junction (NS-J) spectra can then be analysed based on microscopic considerations (Supplementary Note 5).

Equation (3) relates the DC resistivity of a nanosheet network, $\rho_{Net}$, to the resistance of the average nanosheet-junction pair, $(R_{NS} + R_J)$(Supplementary Note 1, Supplementary Sections 1.1 and 1.3). We propose that the same scaling exists between the complex resistivity of the network, $\rho^*_{Net}$, and $Z_{NS-J}$ (concept and derivation in Supplementary Notes 5 and 6). Here, $\rho^*_{Net} = Z_{Net}A_{Net}/L_{Ch}$, where $A_{Net}$ and $L_{Ch}$ are the network cross-sectional area and channel length. This yields an equation which converts the real and imaginary parts of $\rho^*_{Net}$ to those representing the average nanosheet-junction pair, once $P_{Net}$, $t_{NS}$, $l_{NS}$, and $n_{NS}$ are known (although when $n_{NS}$ is large enough the square-bracketed term can be neglected):

$$Z_{NS-J} = \rho^*_{Net} \frac{(1 - P_{Net})}{2t_{NS}} \left[ 1 + \frac{2}{n_{NS}t_{NS}l_{NS}^2} \right]^{-1} \quad (4)$$

We demonstrate this impedance approach using liquid-deposited networks of electrochemically exfoliated MoS₂ nanosheets ($l_{NS} \approx 1\,\mu m$, $t_{NS} \approx 3.3\,nm$) with low porosity[47] and large-area junctions[19] (Fig. 4a). While this is an intensively studied system due to its relatively high mobility (>1 cm² V⁻¹ s⁻¹ for printed networks)[19–21,48], the actual $R_{NS}$ and $R_J$ values are completely unknown. We first measure the (peak) field-effect mobility of these networks in a transistor geometry (Fig. 4b and Supplementary Note 7), obtaining an average of $\mu_{Net} = (6.6 \pm 0.6)$ cm² V⁻¹ s⁻¹, consistent with previous measurements[19].

We then measured the real (Re) and imaginary (Im) parts of the complex network resistivity as a function of frequency, $\omega$, as shown in Fig. 4c. As the features of relevance occur at frequencies >10 kHz, it is essential that background artefacts, such as stray capacitances and inductances, are minimised (Supplementary Notes 8–10), and the potential influence of contact resistance is accounted for (Supplementary Note 11). The low frequency plateau of Re($\rho^*_{Net}$) in Fig. 4c yields a DC resistivity of $\rho_{Net} = 0.024$ Ω m. Combining this value with the measured mobility gives a carrier density for this network of $3.8 \times 10^{23}$ m⁻³, close to previously reported values for electrochemically exfoliated MoS₂[21,49].

With these values now known, Eq. (4) can be used to convert the network impedance, $Z_{Net}$, into the impedance of the average nanosheet-junction pair, $Z_{NS-J}$ (Supplementary Notes 5 and 6). The real component of $Z_{NS-J}$ is shown in Fig. 4d, with the inset showing the imaginary component. As shown in Supplementary Note 11, we found $Z_{NS-J}$ to be independent of channel length, which allows us to rule out the effects of contact resistance.

In the AC domain, the nanosheet-junction pair can be described as the nanosheet resistance, $R_{NS}$, in series with a parallel resistor, $R_J$, and capacitor, $C_J$, representing the junction (Fig. 4c, inset), an arrangement referred to as the Randles circuit (Supplementary Note 12). We chose to fit the Re($Z_{NS-J}$) spectrum as the extracted parameters have a higher accuracy compared to fitting other spectra (Supplementary Note 13). Such spectra can be fitted using equations appropriate to the Randles circuit to yield values of $R_{NS}$, $R_J$, and $C_J$. We account for the distribution of junction resistances by fitting the data using a modified equation for the Randles circuit[50] (Supplementary Eq. (S8), Supplementary Note 12):

$$\mathrm{Re}Z_{NS-J}(\omega) = R_{NS} + \frac{R_J[1 + (\omega R_J C_J)^n \cos(n\pi/2)]}{1 + 2(\omega R_J C_J)^n \cos(n\pi/2) + (\omega R_J C_J)^{2n}} \quad (5)$$

where $n$ is an ideality factor that decreases from 1 as the distribution of $R_J C_J$ values in the network broadens (see ref. 51 and Supplementary Note 12).

We find Eq. (5) to fit our data very well, yielding values of $R_J = (2.9 \pm 0.1)$ MΩ, $R_{NS} = (0.67 \pm 0.07)$ MΩ, $C_J = (8.4 \pm 0.4) \times 10^{-15}$ F, and $n \approx 0.985$. Over five devices on the same substrate, $R_{NS}$ typically varies by <20%, with $R_J$ and $C_J$ showing wider distributions (with a standard deviation/mean of <60%) due to spatial morphology variations (Supplementary Note 14). Here $R_J$ is >1000× lower than in Fig. 3d, e for LPE nanosheets, while $R_J/R_{NS} = 4.4 \pm 0.2$, meaning it is much less junction-limited than the LPE WS₂ and WSe₂ networks presented above. We can further analyse the nanosheet resistance by converting it to nanosheet resistivity using $\rho_{NS} = 2R_{NS}t_{NS}$ (or directly from the network impedance spectrum as described in Supplementary Note 15), obtaining $\rho_{NS} = (4.4 \pm 0.5) \times 10^{-3}$ Ω m. As the meso-porosities of networks of electrochemically exfoliated nanosheets are very low (≈0.02)[19,47], we make the assumption that the average number of carriers per volume of network is the same as the average number of carriers per volume of nanosheet ($n_{Net} \approx n_{NS}$)[52]. This allows us to calculate a nanosheet mobility of $37 \pm 4$ cm² V⁻¹ s⁻¹, reasonable for electrochemically exfoliated MoS₂[21,53].

We can support this result using several direct measurements. First, we used time-resolved pump-probe terahertz spectroscopy to determine the room-temperature AC mobility of photogenerated charge carriers (Supplementary Note 16). The observed mobility at a frequency of 1 THz is $40 \pm 2$ cm² V⁻¹ s⁻¹, consistent with the value implied by impedance. Second, we performed field-effect mobility

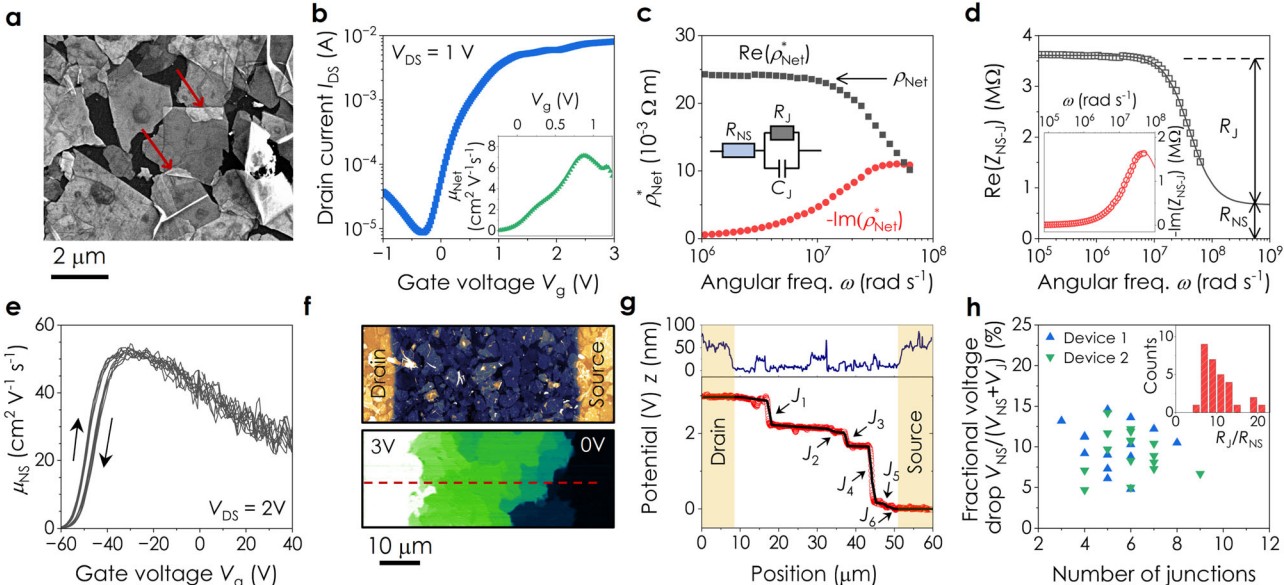

**Fig. 4 | Identification of nanosheet and junction resistances. a** SEM surface image of a network of electrochemically exfoliated (EE) MoS$_2$ nanosheets. The arrows point to two well-defined junctions. **b** Field effect transfer curve for an electrolytically gated EE MoS$_2$ network using a drain-source voltage of $V_{DS} = 1$ V. Inset: Plot of the network mobility, $\mu_{Net}$, as a function of gate voltage. Averaging over four devices yields a mean (peak) mobility of $\mu_{Net} = (6.6 \pm 0.6)$ cm$^2$ V$^{-1}$ s$^{-1}$. **c** Real (Re) and imaginary (Im) parts of the complex network resistivity, $\rho^*_{Net}$, plotted as a function of angular frequency, $\omega$, for a network of EE MoS$_2$ nanosheets. Inset: The circuit element representing a nanosheet-junction pair. Here, $R_{NS}$ is the nanosheet resistance while $R_J$ and $C_J$ are the junction resistance and capacitance respectively. **d** The real part of the impedance of a nanosheet-junction pair, Re($Z_{NS-J}$), plotted versus $\omega$. The data has been fitted using Eq. (5) and the contributions of the junction and nanosheet resistances are indicated by the arrows. Inset: -Im($Z_{NS-J}$) plotted as a function of $\omega$. The solid line is a fit, see Supplementary Note 13 for equation and fit

parameters. **e** Gate-voltage-dependent mobility, $\mu_{NS}$, for a representative individual EE MoS$_2$ nanosheet. Arrows indicate the sweep direction. **f** Topographic AFM image (top) and in-operando KPFM image (bottom) of a section of an EE MoS$_2$ network between source and drain electrodes. **g** Topographic line profile (top) and potential profile (bottom) associated with the red dashed line in **f**. In this section of channel, 6 sharp drops associated with inter-sheet junctions can be seen, labelled as $J_1$ to $J_6$. The nearly flat regions represent the gradual drop of potential across nanosheets. The black line represents fits to the linear regions. **h** Fractional voltage dropped across nanosheets in a given portion of channel plotted versus the number of junctions observed in that section. The fractional voltage drop is given by $V_{NS}/(V_{NS} + V_J)$ where $V_{NS}$ and $V_J$ describe voltage drops across nanosheets and junctions respectively. Inset: Histogram of $R_J/R_{NS}$ values calculated from the fractional voltage drops in **h**.

measurements on individual MoS$_2$ nanosheets (see Fig. 4e and Supplementary Note 17) obtaining a zero-gate-bias value of $42 \pm 6$ cm$^2$ V$^{-1}$ s$^{-1}$, again consistent with our results. Combining this value with the $\mu_{Net}$ value extracted using impedance spectroscopy, and reformulating Eq. (1) as $(R_J/R_{NS}) \approx (\mu_{NS}/\mu_{Net}) - 1$ (neglecting the final term as $n_{NS}$ is large), we can estimate $R_J/R_{NS} = 5.3 \pm 1.4$, again within error of the impedance result.

Finally, we used in-operando frequency-modulated Kelvin probe force microscopy (KPFM) measurements to map out the spatial distribution of the electrostatic potential across an MoS$_2$ network (Fig. 4f, g)[54–56]. Between the biased and grounded electrodes, we find a combination of gradual decreases in potential within the nanosheets and well-defined potential drops at the junctions (Fig. 4g). By summing the potential drops at the junctions along the channel length, we extract the overall fraction of potential dropped within the nanosheets, which yields a mean value of $R_J/R_{NS} = 10 \pm 4$ (Fig. 4h). Although microstructural variations in similarly deposited networks will cause differences in $R_J$, we find these data to be highly consistent, supporting the validity of the impedance method. Furthermore, to demonstrate that the impedance technique can be applied to characterise nanosheet networks beyond MoS$_2$, we show preliminary data for liquid-deposited networks of electrochemically exfoliated MoSe$_2$ and Nb-doped MoSe$_2$ in Supplementary Note 18.

**Using the impedance method: temperature dependence**

Impedance spectroscopy allows $R_J$ and $R_{NS}$ to be measured simultaneously under various circumstances. We demonstrate this by performing impedance measurements on networks of electrochemically exfoliated MoS$_2$ at various temperatures (Fig. 5). The low frequency

limit of the Re($\rho^*_{Net}$) spectrum (Fig. 5a) yields the DC network resistivity ($\rho_{Net}$) which is plotted versus $1/T$ in Fig. 5b. Previous measurements on electrochemically exfoliated MoS$_2$ networks have shown $\rho_{Net}$ to follow activated behaviour around room temperature ($\rho_{Net} = \rho_0 \exp(E_a/k_B T)$, $\rho_0$ and $E_a$ are constants) but 3D variable-range hopping[57] (3D-VRH) at lower temperatures ($\rho_{Net} = \rho_0 \exp[(T_0/T)^{1/4}]$, $\rho_0$ and $T_0$ are constants)[58]. As shown in Fig. 5b and its inset, our data is consistent with this behaviour (with fit constants in-panel). However, this standard analysis cannot distinguish the respective contributions from the nanosheets and junctions. To decouple these properties, we first convert the network impedance spectra to Re($Z_{NS-J}$) and -Im($Z_{NS-J}$) spectra (Fig. 5c, d), obtaining spectra which display a well-defined temperature dependence.

Fitting the Re($Z_{NS-J}$) spectrum to Eq. (5) yields values of $R_{NS}$, $R_J$, and $C_J$ for all temperatures, as shown in Fig. 5e, f (see Supplementary Note 19 for further detail including fitting the Im($Z_{NS-J}$) spectra). Opposing temperature dependences for $R_J$ and $R_{NS}$ (Fig. 5e) indicate hopping and band-like transport, respectively, with $R_{NS}/R_J$ increasing with temperature. Figure 5f shows a relatively small change in the junction capacitance, $C_J$, over the temperature range, meaning the primary changes in the Re($Z_{NS-J}$) spectrum are associated with $R_J$.

We find typical $C_J$ values of 6–8 fF which, combined with SEM measurements of junction area where $A_J = 0.4$ μm$^2$ (Fig. 5f, inset, and Supplementary Note 20), give $C_J/A_J \approx 2$ μFcm$^{-2}$. This is considerably smaller than typical quantum capacitances ($\approx 10$ μFcm$^{-2}$)[59] but consistent with a geometric capacitance described by $C_J/A_J = \varepsilon_r \varepsilon_0/l_J$. By taking $\varepsilon_r = 1$ and an inter-sheet distance of $l_J = 0.6$ nm[21], we find $C_J/A_J = 1.5$ μFcm$^{-2}$, similar to the measured value. This allows us to use the model described in Supplementary Note 21 to estimate the

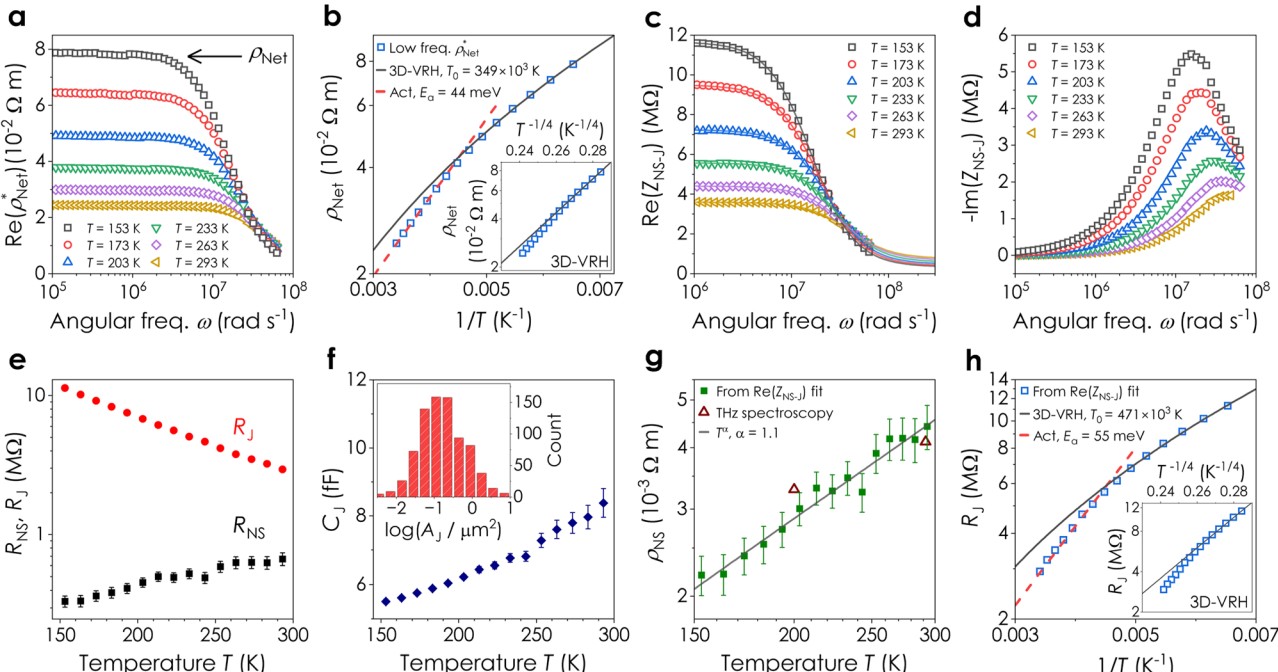

**Fig. 5 | Measurement of network, nanosheet, and junction transport using impedance spectroscopy. a** Real part of the complex network resistivity, $\mathrm{Re}(\rho^*_\mathrm{Net})$, plotted versus angular frequency, $\omega$, for a network of EE MoS$_2$ nanosheets at a range of temperatures, $T$. The arrow indicates that the DC network resistivity was found from $\rho_\mathrm{Net} = \mathrm{Re}(\rho^*_\mathrm{Net})_{\omega\to 0}$. **b** DC network resistivity, $\rho_\mathrm{Net}$, plotted as a function of temperature as $1/T$ and $T^{-1/4}$ (inset). The dashed line is an activated (Act) fit while the solid line is a fit to the 3D variable-range hopping (VRH) model. Real (**c**) and imaginary (**d**) parts of the impedance spectrum of a single (average) nanosheet junction pair, $Z_\mathrm{NS\text{-}J}$, measured at various temperatures. The curves in **c** are fitted using Eq. (5). **e** Nanosheet and junction resistances, $R_\mathrm{NS}$ and $R_\mathrm{J}$, extracted from fits to the $\mathrm{Re}(Z_\mathrm{NS\text{-}J})$ spectra, plotted as a function of temperature. The uncertainty in $R_\mathrm{NS}$ and $R_\mathrm{J}$ is ±the error in the fit. **f** Junction capacitance, $C_\mathrm{J}$, plotted versus temperature. Inset:

Histogram of nanosheet junction areas, $A_\mathrm{J}$, measured from SEM images and plotted as $\log(A_\mathrm{J}/\mu\mathrm{m}^2)$. This distribution showed $\langle A_\mathrm{J}\rangle = 0.41\,\mu\mathrm{m}^2$ ($n = 807$). The uncertainty in $C_\mathrm{J}$ is ±the error in the fit to the $\mathrm{Re}(Z_\mathrm{NS\text{-}J})$ spectra. **g** Resistivity of an (average) individual nanosheet, $\rho_\mathrm{NS}$, extracted from $R_\mathrm{NS}$ ($\rho_\mathrm{NS} \approx 2R_\mathrm{NS}t_\mathrm{NS}$) and plotted as function of temperature. The solid line is a power law with exponent $\alpha = 1.1$. The uncertainty in $\rho_\mathrm{NS}$ is ±the RSS of SE in the mean for $t_\mathrm{NS}$ ($n = 674$) and the error in the fit for $R_\mathrm{NS}$. The hollow triangles represent the THz mobility of the nanosheets converted into resistivity using the measured carrier density of $3.8 \times 10^{23}\,\mathrm{m}^{-3}$. **h** Junction resistance plotted as a function of $1/T$ and $T^{-1/4}$ (inset). The dashed line is an activated-behaviour fit (Act) described by Eq. (6), while the solid line is a fit to the 3D-VRH model (Eq. (7)). The uncertainty in $R_\mathrm{J}$ is ±the error in the fit to the $\mathrm{Re}(Z_\mathrm{NS\text{-}J})$ spectra in **c**.

effective permittivity of the network, finding a value $>10^4$, in agreement with the measured network capacitance.

To separately assess the transport mechanisms associated with the nanosheet and the junction, we examine the temperature dependence of $\rho_\mathrm{NS}$ and $R_\mathrm{J}$ (Fig. 5g, h). Figure 5g shows $\rho_\mathrm{NS}$ to scale as a power law ($\rho_\mathrm{NS} \propto T^\alpha$), with $\alpha \approx 1.1$, consistent with measurements on individual MoS$_2$ nanosheets (typically $\alpha = 0.5\text{–}1.9$)[60–62]. This behaviour implies band-like transport, limited by phonon scattering[63], which is commonly seen for individual MoS$_2$ nanosheets with high carrier densities[62,64,65], and is also in agreement with the THz spectroscopy data (Fig. 5g, triangles).

As these networks are junction-limited, the temperature dependence of $R_\mathrm{J}$ in Fig. 5h is similar to that of $\rho_\mathrm{Net}$, showing the same transition from variable-range hopping to activated behaviour. We propose this behaviour is consistent with Miller-Abrahams-type[57] hopping between nanosheets such that:

$$R_\mathrm{J} \approx R_\mathrm{J,0}\exp(2l_\mathrm{J}/a)\exp(E_\mathrm{a}/k_\mathrm{B}T) \tag{6}$$

where $R_\mathrm{J,0}$ is a constant, $a$ is the localisation length and $E_\mathrm{a}$ is the activation energy. In Supplementary Note 22, we derive an alternative version of the 3D-VRH model, considering inter-nanosheet hopping from the conduction band-edge of one nanosheet to the conduction band-edge of another yielding:

$$R_\mathrm{J} \approx R_\mathrm{J,0}\exp\left(\frac{2l_\mathrm{J}}{a}\right)\exp\left(\left[\frac{T_0}{T}\right]^{1/4}\right) \tag{7}$$

where the constant $T_0$ is given by $T_0 \sim 76\pi\hbar^2 d_0/k_\mathrm{B}a^3m$, with $d_0$ being the monolayer thickness and $m$ is the effective electron mass. Fitting the data in Fig. 5h to Eq. (6) at higher temperatures and Eq. (7) at lower temperatures yields $E_\mathrm{a} = 55 \pm 2$ meV and $T_0 = (471 \pm 37) \times 10^3$ K, values which are solely associated with the junctions. Our $E_\mathrm{a}$ value is smaller than other reported values (in the absence of gating[58]), which is consistent with our low $R_\mathrm{J}$ (Eq. (6)) and relatively high network carrier mobility[66]. Combining $T_0$ with $m = 0.7m_\mathrm{e}$ and $d_0 = 0.6$ nm, we calculate $a = 0.7$ nm, similar to published values for MoS$_2$ (0.2–3 nm)[58,67–69]. The most probable hopping distance was $\approx 2$ nm, again consistent with inter-sheet hopping (Supplementary Note 22).

## Discussion

Our simple model for conduction in nanoparticle networks is highly useful for describing the resistivity of printed networks for a range of nanomaterials. It naturally explains counterintuitive behaviour such as the increase in network resistivity with the size of conducting nanosheets and the non-monotonic dependence of network resistivity on semiconducting nanosheet size. The model enables data fitting, allowing the junction and particle resistances to be extracted from DC electrical measurements. The resultant data confirms printed networks to be junction limited and provides insights into the magnitude of junction resistances and the relationship between $R_\mathrm{J}$ and intrinsic nanosheet properties such as $\rho_\mathrm{NS}$. In addition, the model directly enables AC impedance spectroscopy to be used to measure $R_\mathrm{J}$ and $\rho_\mathrm{NS}$ in a single measurement, allowing one to study both inter- and intra-nanosheet transport mechanisms simultaneously. We believe this

work supplies a valuable tool for analysis of printed networks of technologically important nanomaterials.

## Methods

### Ink preparation – liquid-phase exfoliation (LPE)

Graphene, $WS_2$ and $WSe_2$ nanosheets were produced by horn probe sonication (Sonics Vibra-cell VCX-750 ultrasonic processor) of bulk powders[70]. Graphite (Asbury Carbons, grade 3763) and $WSe_2$ (10–20 μm, 99.8% metals basis, Alfa Aesar) powders were first ultra-sonicated in deionised water (DI, 18.2 MΩ, produced in-house) for 1 h at a concentration of 35 mg mL$^{-1}$, with an amplitude of 55% and a pulse rate of 6 s on and 2 s off. The process temperature was maintained at 7 °C using a chiller to prevent overheating of the ultrasonic probe. The resulting dispersions were centrifuged (Hettich Mikro 220R) for 1 h at 2684 × $g$ to remove contaminants from the starting material[71]. The supernatant was decanted, and the sediment was redispersed in 80 mL of DI water and sodium cholate (SC, >99%, Sigma Aldrich) at a concentration of 2 mg mL$^{-1}$. The resulting dispersion was sonicated for 8 h, with a 4 s on and 4 s off pulse rate at an amplitude of 50%. The $WS_2$ nanosheets were produced in a similar manner from commercially sourced bulk powders (10–20 μm, 99.8% metals basis, Alfa Aesar). However, the ultrasonication was carried out using isopropanol (IPA, HPLC grade, Sigma Aldrich) as the solvent.

The stock dispersions produced by liquid phase exfoliation (LPE) of the graphite, $WSe_2$ and $WS_2$ powders were size-selected using liquid cascade centrifugation (LCC)[72]. Here, a polydisperse parent dispersion is separated into fractions of progressively smaller nanosheets by isolating the sediment at well-defined intervals as the relative centrifugal force is increased. These sediments contain the desired nanosheet fractions, which can then be redispersed in solvents as required. Each stock dispersion was first centrifuged at 28 × $g$ for 2 h to remove any unexfoliated material. For graphene, the supernatant was centrifuged at 112 × $g$, 252 × $g$, 447 × $g$, 699 × $g$ and 1789 × $g$ for 2 h. After each step the sediment was retained and redispersed in a reduced volume of fresh DI:SC solution (2 mg mL$^{-1}$) to create a size-selected ink. The fraction captured at 112 × $g$ was subjected to an additional centrifugation step at 28 g for 1 h to generate a further size fraction. The $WSe_2$ parent dispersion was size-selected in the same manner with upper limits of 112 × $g$, 252 × $g$, 447 × $g$, 699 × $g$, 1006 × $g$, 1789 × $g$, 3382 × $g$ and 11,180 × $g$. As with the graphene, the fraction captured at 112 × $g$ was centrifuged at 28 × $g$ for 1 h to generate an additional size fraction. The $WS_2$ stock dispersion was fractionated using upper limits of 112 × $g$, 252 × $g$, 342 × $g$, 699 × $g$, 1006 × $g$, 1789 × $g$ and 4025 × $g$. Here, the largest size was split into 3 fractions by additional centrifugation steps at 28 × $g$ and 63 × $g$ for 1 h. The smallest of these sizes (63 × $g$) was not used.

The size-selected graphene and $WSe_2$ inks were then transferred (by redispersing the sediment) into IPA for spray coating. To ensure that the nanosheets in each fraction were confined to the sediment, samples isolated below 1066 × $g$ were centrifuged at 4052 × $g$ for 2 h. The DI:SC supernatant was discarded, and the sediment was redispersed in IPA. This step was repeated twice to ensure removal of the surfactant. For nanosheet fractions isolated above 1066 × $g$ a RCF of 25,155 × $g$ was used for the transfer steps

Silver nanowire inks (AgNW, A40, 40 nm × 35 μm in IPA, Novarials Corporation) were size-selected using sonication induced scission in an ultrasonic bath. In each case a stock AgNW dispersion (0 h) was sonicated for a fixed duration at a concentration of 1 mg mL$^{-1}$ in IPA. Sonication times of 0.05, 0.25, 0.5, 1, 1.5 and 2 h were used to produce the size-selected AgNW inks.

Size-selected silver nanosheet (AgNS) inks were prepared from commercially sourced stock dispersions (N300 nanoflake and M13 nanoflake, Tokusen Nano). Each stock dispersion was first diluted to a concentration of 100 mg mL$^{-1}$ in DI water. The stock containing the larger nanosheets (M13) was centrifuged at 28 × $g$ for 5 min to remove large material. The supernatant was subjected to a further step at 63 × $g$ for 5 min and the sediment was retained and dispersed in a reduced volume of DI water. The stock of smaller nanosheets (N300) was also centrifuged at 112 × $g$ for 5 min to remove the largest material. This was followed by steps at 447 × $g$, 1006 × $g$, 1789 × $g$, 4025 × $g$ for 5 min each. The sediment at each interval was redispersed in a reduced volume of DI water to create a set of size-selected AgNS inks.

### Ink preparation – electrochemical exfoliation (EE)

Nanosheet inks produced using electrochemical exfoliation were prepared using bulk crystals. $MoS_2$ of natural origin was collected in Krupka, Czech Republic. $MoSe_2$ and Nb-doped $MoSe_2$ were prepared by direct reaction from the elements and subsequent vapour transport by chlorine in a two-zone furnace. Molybdenum (99.999%, -100 mesh, Shanghai Quken New Material Technology Co., China), selenium (99.9999%, granules 2–6 mm, Wuhan Xinrong New Material Co., China), niobium (+99.9%, −100 mesh, Beijing Metallurgy and Materials Technology Co., China), and selenium tetrachloride (99.9%) were used for synthesis.

For synthesis and subsequent crystal growth molybdenum and selenium were placed in an ampoule (250 mm × 50 mm) in a stochiometric amount corresponding to 50 g of $MoSe_2$ together with 0.6 g of $SeCl_4$ and 2 at% excess of selenium inside a glovebox and melt-sealed under high vacuum ($<1 \times 10^{-3}$ Pa). For Nb-doped samples the stochiometric amount of element corresponding to $Mo_{0.97}Nb_{0.03}Se_2$ together with 0.6 g of $SeCl_4$, and 2 at% excess of selenium were placed in an ampoule (250 × 50 mm) inside a glovebox and melt-sealed under high vacuum ($<1 \times 10^{-3}$ Pa). The ampoules were placed in a horizontal muffle furnace and first heated at 500 °C for 25 h, then 600 °C for 50 h, finally at 800 °C for 50 h. The heating and cooling rate was 1 °C min$^{-1}$. Between each heating step, the ampoule was mechanically homogenised for 5 min. The reacted powder in the ampoule was subsequently placed in a two-zone horizontal furnace. First, the growth zone was heated at 1000 °C and the source zone was kept at 800 °C for two days. Next, the thermal gradient was reversed and the source zone was set at 1000 °C with the growth zone at 950 °C. Over a period of 166 h, the temperature of the source zone was increased to 1100 °C while keeping growth zone temperature constant. After 166 h, the thermal gradient was kept constant for another 166 h. Finally, the ampoule was cooled over a period of 4 h at 100 °C in the source zone and 400 °C in the growth zone before the heating was switched off. The ampoule was opened in an argon-filled glovebox and crystals with size up to 4 cm were removed from ampoule.

An electrochemical setup consisting of two electrodes was employed to intercalate bulk 2D crystals (cathode), while a platinum foil (Alfa Aesar) served as the anode. The electrolyte solution was prepared by adding tetrapropylammonium (TPA) bromide (Sigma Aldrich, 5 mg mL$^{-1}$) to propylene carbonate (≈50 mL). An 8 V potential difference was applied for 30 min between the electrodes to facilitate the intercalation of the 2D crystal with TPA+ cations. The expanded material was washed with dimethylformamide (DMF, HPLC grade, Sigma Aldrich) to remove residual propylene carbonate and bromine. The 2D crystal was then bath-sonicated in 1 mg mL$^{-1}$ poly(-vinylpyrrolidone) (PVP, molecular weight ≈40000) in DMF for 5 min followed by centrifugation (Hettich Mikro 220 R) at 24 × $g$ for 20 min to remove unexfoliated crystals. The dispersion was size-selected by centrifuging the supernatant (top 90%) at 97 × $g$ for 1 h and collecting the sediment. The sediment was diluted with 2 mL of DMF and centrifuged at 9744 × $g$ for 1 h twice to remove the residual PVP. A third washing step was used to remove residual DMF, which involved redispersing the sediment in IPA (0.5 mL) and subsequently centrifuging at 9744× g for 1 h. The sediment was then redispersed in IPA (≈0.5 mL, concentration ≈2.5 g L$^{-1}$) to make the 2D crystal dispersions used in this study.

## Nanosheet & ink characterisation

Atomic force microscopy (Bruker Multimode 8, ScanAsyst mode, non-contact) was used to measure the nanosheet thickness and lateral dimensions in the graphene, $WS_2$, $MoS_2$ and AgNS inks. Measurements were performed in air under ambient conditions using aluminium coated silicon cantilevers (OLTESPA-R3). The concentrated dispersions were diluted with isopropanol to optical densities <0.1 at 300 nm. A drop of the dilute dispersion (10 μL) was flash-evaporated on pre-heated (175 °C) Si/SiO₂ wafers (300 nm oxide layer, $0.5 \times 0.5\,cm^2$, MicroChemicals). After deposition, the wafers were rinsed with ~10 mL of water and ~10 mL of isopropanol and dried with compressed nitrogen. Typical image sizes ranged from $15 \times 15\,\mu m^2$ for larger nanosheets to $3 \times 3\,\mu m^2$ for small nanosheets at scan rates of 0.4–0.8 Hz with 1024 lines per image. Previously published length corrections were used to correct lateral dimensions from cantilever broadening[73]. Bright-field transmission electron microscopy (TEM) was performed using a JEOL 2100 system operating at an accelerating voltage of 200 kV. Samples were diluted and drop-cast onto holey carbon grids (Agar Scientific) for imaging. The grids were placed on filter membranes to wick away excess solvent and dried overnight at 120 °C in a vacuum oven. The average nanosheet length in each size-selected $WSe_2$ ink was determined by measuring the longest axis of each imaged nanosheet and denoting it as its length. UV-Vis optical spectroscopy (Perkin Elmer 1050 spectrophotometer) was used to determine the concentration of the graphene[74], $WS_2$[73] and $WSe_2$[75] inks using previously reported spectroscopic metrics. Each ink was diluted to a suitable optical density and extinction spectra were recorded in 1 nm increments using a 4 mm quartz cuvette. The AgNW length in each fractionated ink was determined by drop casting 300 μL of ink, diluted to a concentration of 0.01 mg mL⁻¹, onto Au-coated Si/SiO₂ substrates heated to 150 °C and measured from SEM images. The AgNS ink concentration was calculated by vacuum filtration of a known volume of each size-selected ink onto an alumina membrane (Whatman Anodisc, 0.02 μm pore size) and weighing.

## Network deposition

Spray coating was performed using a Harder and Steenbeck Infinity airbrush attached to a computer-controlled Janome JR2300N mobile gantry. All deposited traces were defined using stainless steel shadow masks on substrates heated to a temperature of 80 °C. A $N_2$ back pressure of 45 psi, nozzle diameter of 400 μm and stand-off distance of 100 mm between the nozzle and substrate were used[76]. The size-selected graphene inks were diluted to a concentration of 0.2 mg mL⁻¹ for spraying. The AgNW, $WS_2$ and $WSe_2$ inks were sprayed at a concentration of 0.5 mg mL⁻¹. The above traces were patterned onto ultrasonically cleaned glass slides (VWR). The AgNS inks were deposited at a concentration of 5 mg mL⁻¹ onto Al₂O₃-coated PET substrates (Mitsubishi Paper Mills). Prepatterned gold bottom electrodes (5 nm/95 nm Ti/Au) were deposited onto the glass substrates to facilitate electrical measurements on the sprayed graphene and $WS_2$ networks using a Temescal FC2000 metal evaporation system. Inter-digitated (IDE) silver nanoparticle (<50 nm diameter, 30–35 wt% in methyltriglycol, Sigma Aldrich) top electrodes were aerosol jet printed onto the $WSe_2$ networks (Optomec AJP300).

For the Langmuir Schaefer-type (LS) deposition a custom-built setup was used, as published recently[19,77]. Fused silica (MicroChemicals), Si/SiO₂ (300 nm oxide layer, MicroChemicals), and microscope slide (VWR) substrates were first pretreated with KOH to remove surface contaminants and etch the surface to promote nanosheet adhesion. A 250 mL beaker was then filled with high-purity water until the substrate on the substrate holder was completely submerged. Approximately 2 mL of distilled *n*-Hexane (HPLC grade, Sigma Aldrich) was introduced into the water in the beaker to establish the liquid/liquid interface. Using a Pasteur pipette, the nanosheet ink was then carefully injected into the interface until a uniform film was observed. Subsequently, the substrate was lifted through the liquid/liquid interface to transfer the nanosheet layer. The wet substrate was allowed to air dry at room temperature. To eliminate any remaining water from nanosheet junctions and interfaces, dry films were annealed at 120 °C for 2 h under an argon atmosphere before further depositions or characterisation.

## Network characterisation

Scanning electron microscopy (SEM) of the deposited nanosheet and nanowire networks was performed using a Carl ZEISS Ultra Plus SEM. Samples were mounted on aluminium SEM stubs using conductive carbon tabs (Ted Pella) and grounded using conductive silver paint (PELCO, Ted Pella). All images were captured at an accelerating voltage of 2 kV using a working distance of 5 mm and a 30 μm aperture. Both the Inlens and SE2 detectors were used for imaging. The thickness of the deposited networks was determined using a combination of contact ($WSe_2$, graphene and AgNSs) and optical ($WS_2$) profilometry, as well as from SEM cross-sections (AgNWs) and AFM (LS films). Contact profilometry was performed using a Bruker Dektak stylus profilometer (10 μm probe, 19.6 μN force). An optical profilometer (Profilm3D, Filmetrics) operating in white-light interferometry mode with a 50× objective lens was used for non-contact thickness measurements.

## DC electrical characterisation

Direct current (DC) electrical characterisation of the printed networks was performed in ambient conditions using a Keithley 2612 A sourcemeter connected to a probe station. Two-terminal measurements in an interdigitated electrode geometry were used to measure the resistance of the printed $WS_2$ ($L_{Ch} = 50\,\mu m$, $W_{Ch} = 19.4\,mm$) and $WSe_2$ networks ($L_{Ch} = 85\,\mu m$, $W_{Ch} = 4.3\,mm$). Prepatterned electrodes were used to characterise the printed graphene networks using two-terminal measurements in a transmission line geometry ($L_{Ch} = 1.4$–$20.2\,mm$, $W_{Ch} = 1\,mm$). Four-terminal measurements were used to determine the resistance of the printed AgNS ($L_{Ch} = 3\,mm$, $W_{Ch} = 1\,mm$) and AgNW ($L_{Ch} = 35.5\,mm$, $W_{Ch} = 500\,\mu m$) networks. Evenly spaced electrical contacts were painted onto the samples using conductive silver paint (PELCO, Ted Pella).

## AC electrical characterisation

Impedance spectra were taken using a Keysight E4990E analyser with a 30 MHz maximum frequency. A test fixture (16047E) was used to connect the samples to the analyser as this allowed as short a wire distance as possible (down to 5 cm) to avoid inductive artefacts at high frequency. A spring-loaded probe attachment (Sensepeek SP10) was used to connect the analyser to the contact pads on the substrates. Ti/Au (5 nm/95 nm) electrodes were deposited by evaporation (FC-2000 Temescal Evaporator) through a shadow mask ($L_{Ch} = 50\,\mu m$, $W_{Ch} = 19.4\,mm$) for AC electrical characterisation. For contact resistance measurements, electrodes with five different channel lengths ($L_{Ch} = 50, 80, 100, 150$ and $200\,\mu m$, $W_{Ch} = 19.4\,mm$) were used. The spectra were acquired with a 500 mV amplitude using a precision speed of 3. A DC voltage sweep was first run on the sample to ensure the response is linear through the origin in the range of the AC amplitude.

Temperature-dependent impedance measurements were performed using a broadband Alpha High-Resolution Impedance Analyser (Novocontrol GmbH, Germany), which utilizes a capacitance bridge technique to calculate impedance. The real and imaginary components of impedance were measured from a frequency of 100 Hz to 10 MHz in the temperature range 20 °C to −120 °C. The samples were placed inside a sample holder which has a fitted Pt 100 Ω resistance temperature sensor in contact with the electrodes. The temperature of the sample was controlled inside a double wall cryostat and maintained by a heated $N_2$ jet produced by evaporating liquid nitrogen inside a 50 L dewar (Apollo 50, Messer Griesheim GmbH). The Quatro

temperature controller controls the power supplied to the dewar and gas heater. The AC measuring voltage applied to the sample was set at 0.1 V.

## Terahertz (THz) spectroscopy

The intrinsic mobility of charge carriers was determined from optical-pump terahertz-probe (OPTP) and time-resolved THz spectroscopy (TRTS) measurements, as described previously[78–80]. The THz spectroscopy setup used is based on a titanium-doped regenerative amplifier (Libra), producing 60 fs laser pulses with a centre wavelength of 800 nm. The output of the amplifier is split into three parts: (1) optical photoexcitation of the sample (pump), (2) THz generation, and (3) THz detection. The first part of the beam is optically converted to a pump wavelength of 400 nm (photon energy 3.1 eV) in a BBO crystal via frequency doubling. The second part is used for generation of a THz waveform with a duration of $\approx$1 ps in a nonlinear ZnTe crystal via optical rectification. The third part is used for detection of the THz waveform after transmission through the sample, which occurs in another ZnTe crystal via electro-optic sampling. Time delays between the photoexcitation pump pulse and the THz detection pulse ($\tau$) and between the THz generation and detection pulse ($t$) are controlled by mechanical delay stages. All the measurements were performed in a closed box under an $N_2$ atmosphere, and a closed cycle He-cryostat was used for obtaining low temperature data. The time-dependent transmitted THz waveform of the sample without photoexcitation, $E^{off}(t)$, was first measured by so-called THz time-domain spectroscopy (THz-TDS).

During the OPTP measurements, the sample was photoexcited with chopped pump laser pulses of 3.1 eV photons to obtain the difference, $\Delta E(\tau)$, of the maximum of the transmitted THz waveform at a delay $\tau$ after the pump pulse. Hence, $\Delta E(\tau) = E^{off}(t_{max}) - E^{on}(t_{max}, \tau)$, where $t_{max}$ is the time at which the THz waveform is maximum without photoexcitation of the sample. From these measurements we can determine the real part of the photoconductivity averaged over the frequencies in the THz waveform, provided the phase shift of the THz waveform due to the imaginary photoconductivity is negligible[80]. The sum of the products of the quantum yields of electrons and holes ($\Phi_{e,h}(\tau)$) and their respective mobilities ($\mu_{e,h}$) at time $\tau$ after the pump pulse were obtained according to

$$S(\tau) = \Phi_e(\tau)\mu_e + \Phi_h(\tau)\mu_h = \frac{\varepsilon_0 c(n_f + n_b)}{eN_a}\left[\frac{\Delta E(\tau)}{E^{off}(t_{max})}\right]$$

In the equation above, $N_a$ is photoexcitation density per unit area ($2.7 \times 10^{12}$ photons cm$^{-2}$), $\varepsilon_0$ is the vacuum permittivity, $c$ is the speed of light, while $n_f$ and $n_b$ are the refractive indices of the media in front and back of the sample, respectively. Here, we studied films of $MoS_2$ deposited on a quartz substrate. Therefore, in the equation above we used $n_f = 1$ (for $N_2$) and $n_b = 2$ (for the quartz substrate)[81].

For the TRTS measurements we measured the change of the THz waveform at time $\tau = 5$ ps after photoexcitation of the sample by chopping the pump laser pulse and scanning the delay time ($t$) of the THz generation pulse. Together with $E^{off}(t)$ from the THz-TDS measurement we obtain the frequency dependent THz conductivity according to

$$S(\omega, \tau) = \Phi_e(\tau)\mu_e(\omega) + \Phi_h(\tau)\mu_h(\omega) = \frac{c\varepsilon_0(n_f + n_b)}{eN_a}\left[\frac{E^{off}(\omega) - E^{on}(\omega, \tau)}{E^{on}(\omega, \tau)}\right]$$

with $E^{off}(\omega)$ and $E^{on}(\omega, t)$ being the Fourier transforms of the THz waveforms at radian frequency $\omega = 2\pi f$.

## Kelvin probe force microscopy (KPFM)

In-operando KPFM experiments were performed on the AIST NT scanning probe microscopy system under ambient conditions and in a frequency modulated regime. The contact potential difference (CPD) maps were recorded in a two-pass mode, using lift height of 20 nm. Potential drop maps were extracted from the CPD maps by subtracting the reference grounded measurement of the same area, following the procedure described in refs. 56,82. Nu-Nano SPARK probes were used with a Pt coating, spring constant of $\approx$42 N m$^{-1}$, and tip radius below 30 nm. The external bias was provided via a custom-built electrical holder and by using a Keithley 2636 A dual source metre. The ground of the KPFM probe was connected also to the ground of the device (source electrode).

## Transistor measurements on a nanosheet network

After a single Langmuir–Schaefer deposition the $MoS_2$ networks had a film thickness of ~15 nm. Interdigitated electrodes (Ti/Au, 5 nm/ 95 nm) were then deposited (FC-2000 Temescal Evaporator) through a shadow mask ($L_{Ch} = 50$ μm, $W_{Ch} = 19.4$ mm) onto the sample. The ionic liquid 1-ethyl-3-methylimidazolium bis(trifluoromethylsulfonyl)imide (EMIM-TFSI, 98 %, HPLC, Sigma Aldrich) was utilised to regulate ion injection into the semiconducting channel. The ionic liquid was first heated under vacuum at 100 °C for 6 h to degas any absorbed water. Subsequently, a small amount of EMIM-TFSI was carefully pipetted onto the transistor, ensuring the gate and channel were adequately covered. To remove any remaining water, the devices were left in a Janis probe station under vacuum conditions overnight, lasting 12 h. After this step, the devices were returned to atmospheric pressure in preparation for measurements. For electrical characterisation, a Keithley 2612 A dual-channel source measuring unit was used. The transfer characteristics were undertaken within a gate voltage window of −3 to 3 V, employing a scan rate of 50 mV s$^{-1}$. Additionally, $V_{DS}$ was set to 1 V for all the devices during the measurements.

## Transistor measurements on an individual nanosheet

For the electrical measurements of the individual EE $MoS_2$ nanosheet devices a Keithley 2636 A dual source-meter was used with an Instec compact vacuum probe station. The measurements were performed under low vacuum ($10^{-2}$ mbar) and at 300 K. For each device electrical transfer curves ($I_D(V_{SG})$) were measured with varied $V_{SD}$ bias, and apparent linear mobility was extracted by considering the channel geometries and the capacitance of a 300 nm thick global $SiO_2$/Si gate.

## Reporting summary

Further information on research design is available in the Nature Portfolio Reporting Summary linked to this article.

## Data availability

The data supporting the findings of this study are available from the corresponding author upon request. Source data are provided with this paper.

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

## Acknowledgements

We acknowledge funding from the European Union through the ERC grant FUTUREPRINT, the Graphene Flagship and the Horizon Europe project 2D-PRINTABLE (GA-101135196). We have also received support from the Science Foundation Ireland (SFI) funded centre AMBER (SFI/12/RC/2278_P2) and availed of the facilities of the SFI-funded advanced microscopy laboratory (AML), additive research laboratory (ARL) and iCRAG. A.G.K. acknowledges funding from the Marie Skłodowska-Curie Postdoctoral Fellowship "NanoHarvest" (Proposal Number: 101107032). T.C. acknowledges funding from a Marie Skłodowska-Curie Individual Fellowship "MOVE" (grant number 101030735, project number 211395, and award number 16883). L.D appreciates support from Science Foundation Ireland (SFI) (18/EPSRC-CDT/3581). E.Ca appreciates support from the Irish Research Council (IRC) (GOIPG/2020/1051). J.M. acknowledges his Margarita Salas fellowship from the Spanish Ministry of Universities. A.M. acknowledges support from the European Research Council Starting Grant POL_2D_PHYSICS (101075821) and the Austrian Science Fund Y1298-N START Prize. NY is funded by the SFI US-Ireland project (21/US/3788). G.G., S.K. and L.D.A.S. received funding from the Netherlands Organisation for Scientific Research (NWO) in the frame-work of the Materials for sustainability and from the Ministry of Economic Affairs in the framework of the PPP allowance. Z.S. was supported by ERC-CZ program (project LL2101) from Ministry of Education Youth and Sports (MEYS) and acknowledges laser infrastructure from project reg. No. CZ.02.1.01/0.0/0.0/15_003/0000444 financed by the EFRR. We thank Prof. Matthias Moebius for useful discussions.

## Author contributions

J.N.C., C.G. and A.G.K conceived and designed the experiments. C.G., A.G.K, E.Co., L.D and T.C carried out the formal analysis. A.G.K, C.G., L.D., J.M., E.Co., D.O.S, E.Ca., J.B.B., S.L. and Z.S. produced samples for analysis. K.S., J.M. and A.D. performed atomic force microscopy measurements. M.A.A. and A.M. performed Kelvin probe force microscopy and individual nanosheet transistor measurements. N.Y. and J.V. assisted with temperature-dependent impedance. G.G., S.K., and L.D.A.S. performed terahertz spectroscopy. Funding was obtained by J.N.C. The manuscript was written and edited by J.N.C., A.G.K and C.G.

## Competing interests

The authors declare no competing interests.
