## [Peer Review File · Nature Communications]

Understanding how junction resistances impact the conduction mechanism in nano-networksREVIEWER COMMENTS

Reviewer #1 (Remarks to the Author):

I reviewed the manuscript 'Understanding how junction resistances impact the conduction mechanism in nanonetworks' with great interest. The topic is suitable for the journal and the analysis, for the most part, seems well-grounded. I had only two concerns I feel authors should address prior to publication:

1. In the second paragraph of the Introduction section, the authors make an assertion that charge mobility in a single nanoscopic entity is higher than that measured in a network of those nanoscopic entities as 'networks are limited by the junctions between particles, with semiconducting networks particularly junction-limited'. While I agree that junction density contribute to the difference in mobility, is it possible other factors such as defect generation during solution processing affect mobility?

Further, is it possible such processing-induced defects affect the analysis of the measurement results for the semiconducting TMDs in Fig. 3e-f? For example could sonication time required to realize differently sized particles affect R_{np} , thereby affecting measured R_{net} value and contributing to/dominating the observed trend in figs d and e? One way to address this question may be to conduct some simple spectroscopic measurements on the chemically exfoliated TMDs sonicated for different time to produce particles with the indicated particles. The literature suggests solution processing-induced defects are worth considering to validate an important hypothesis presented and evaluated in this work, for example: <https://doi.org/10.1021/nl201874w>

2. In the first paragraph on Page 8, the authors make an assumption: 'the network carrier density is the same as the nanosheet carrier density'. Even though mobility data (linked to carrier density by equation 1 in the manuscript) presented in the paper may suggest this is true, is this assumption valid, especially for measurements of photoexcited electrons when 'all networks were thick enough to be in the bulk-like regime' as stated early in the manuscript? Are there prior works supporting this assertion? At first glance, it seems the literature presents examples of differences semiconducting networks
<https://doi.org/10.1021/jp407765r>

Reviewer #2 (Remarks to the Author):

This is a very thorough investigation on quantifying junction resistance in conductive (semi-conductive) nano-networks.

The authors provide interesting methods to measure the junction resistance in these materials by way of fitting theoretical derivations of the junction's impact on the global resistivity of the samples. This is performed by teasing out the particle resistance and the junction resistance's contribution to overall resistivity depending on temperature and frequency in an EIS setup.

Overall, the contribution of the authors is significant, and very thorough characterizations are provided in the Supplementary Information which is commendable.

Here are my comments and suggestions:

1. On page 3 of the manuscript, the authors mention that previous works exist linking junction resistance to network conductivity but these are too complex for routine utilization. I would like to point out two important things: first, reference 29 utilizes a single equation that only involves physical quantities only and no fitting parameters (the number of junction per wire is known from follow-up work). Second, the present work presents an approximation, which has to be fitted to data first to elucidate fit parameters, therefore cannot be predictive, but only retrospective. Although the derivations are useful, they also rely on a single equation, which does not seem less complex than the mentioned work. Rather than making such sweeping statements about previous

work, I would rather highlight this present work in its effort to characterize size dependence of the resistance and novel approaches to tease out R_j v R_{np} .

2. I am not convinced by the derivation of σ_{net} (Eq S1). It relies on the assumption that the conductive path is effectively going straight through the sample jumping on a particle, a junction, a particle, a junction, etc. However, previous works show clearly that the conduction paths are influenced by R_j/R_{np} and that depending on these ratios, and the presence of multiple junctions per particle, the current will explore as many junctions as possible (parallel) to minimize dissipated power ($i^2 R$). I would urge the authors to set up a simulation of overlapping sheets or wires and check whether S1 scales correctly on a simulated sample.

3. I would be interested in a sensitivity analysis of the estimation of R_j depending on the fitted porosity (results of Fig. 3). How much does R_j depend on the fitting parameters ?

4. This is minor, but I would not call the EIS approach a direct measurement of R_j since it relies on fitting a Randles circuit to the R_j - R_{np} pair, bringing in two assumptions: that it's a Randles circuit, and C_j as a fit parameter. The idea is still however very interesting.

Overall I would try and structure the writing a little more clearly. Sentences such as "Measuring RNP and RJ as described above is time-consuming" on page 6 are unclear, and do not naturally bring the author to understand why a new method is presented to measure R_j .

Reviewer #3 (Remarks to the Author):

This work models the resistance of printed nanomaterial films, finding the nanoparticle and the nanoparticle-to-nanoparticle junction resistance through two different electrical techniques: a D.C. resistivity measurement and an A. C. impedance spectrum. The models used in this work are significant to the printing community to better understand how nanoparticles interact to produce electrically conductive and semiconducting films. This work should be published in Nature Communications after completing the following revisions:

1. How are multiple pathways taken into account in these models? Are you assuming that all of the current is flowing through one pathway? If so, how could this model be expanded to deal with multiple current pathways?
2. What is the distance between the contacts for each material used and does this distance play a factor in the resistance extractions? A table of all of constants used in the D.C. and A.C. model such as the area and thickness measurements as well as the extracted values (such as table S1) should be added to the main text as this information is pertinent to the model.
3. Further discussion comparing and contrasting the different materials and their extracted resistances is needed.
4. Could the A. C. impedance spectrum be applied to conductive material? Is there a reason that this method is specifically used with semiconducting materials? Would the temperature dependence measured for the MoS2 sheets produce similar trends for a conductive material and do the models used for MoS2 apply for these conductive materials?
5. If both the D. C. and A. C. measurements were taken on the same sample, would the junction and nanoparticle resistances fall within the same value?
6. It is stated that the conductivity can be tuned by adjusting material dimensions (in the conclusion), which is not analyzed in depth in this work as there is no comparison between different nanoparticle dimensions to show how well this model fits for each dimension parameter considered. Please revise the statement or complete further work.
7. The supplementary information has a significant amount of information, highlighting how thorough the authors were with this work. However, the main text does not accurately reference or reference enough the findings found in the SI. Please go through and update SI references to be as exact as possible so that people reading this work can easily find important information in the SI, including adding a few words for each SI section referenced. Also, there is information in the SI that is not referred to in the main text. Please refer to all of the SI at least once in the text.
8. The addition of an appendix or section of the SI that has abbreviations defined would help

readers keep track of variables and abbreviated words. Some variables are not defined in the main text (such as nnp), requiring the reader to search through the SI to find its meaning which can be tedious.

9. Change the units of figures 5 c - d to be in units of kelvin rather than Celsius since the other data in this figure is in kelvin.

Response to reviewers

REVIEWER COMMENTS

Reviewer #1 (Remarks to the Author):

I reviewed the manuscript ‘Understanding how junction resistances impact the conduction mechanism in nanonetworks’ with great interest. The topic is suitable for the journal and the analysis, for the most part, seems well-grounded. I had only two concerns I feel authors should address prior to publication:

1. In the second paragraph of the Introduction section, the authors make an assertion that charge mobility in a single nanoscopic entity is higher than that measured in a network of those nanoscopic entities as ‘networks are limited by the junctions between particles, with semiconducting networks particularly junction-limited’. While I agree that junction density contribute to the difference in mobility, is it possible other factors such as defect generation during solution processing affect mobility?

This is in principle possible (although many papers show it is a minimal effect for many processing methods) and so is worth addressing. The main point is that one does not know whether the network mobility is low because junction resistance is high or because nanoparticle resistance is high (i.e. because processing-induced defects have reduced nanoparticle mobility). This underlines our argument that one needs to be able to measure both nanoparticle and junction resistance to determine which scenario is correct. To address this, we have added the following paragraph to the manuscript introduction:

“On the other hand, due to our inability to easily measure either the junction or the nanoparticle resistance *in situ*, even proving that low mobility is due to junction-limitations is challenging. For example, one might argue that the exfoliation process can introduce defects into the nanoparticles which reduces their intrinsic mobility (although in reality this is unlikely, as we argue in Supplementary note 3). This would decrease the network mobility even for a negligible junction resistance. Thus, to fully understand the reason why the network mobility is lower than that of the nanoparticles, one must be able to measure R_J and R_{NP} to determine which is larger and so pinpoint the limiting factor.”

Further, is it possible such processing-induced defects affect the analysis of the measurement results for the semiconducting TMDs in Fig. 3e-f? For example, could sonication time required to realize differently sized particles affect R_{np} , thereby affecting measured R_{net} value and contributing to/dominating the observed trend in figs d and e? One way to address this question may be to conduct some simple spectroscopic measurements on the chemically exfoliated TMDs sonicated for different time to produce particles with the indicated particles. The literature suggests solution processing-induced defects are worth considering to validate an important hypothesis presented and evaluated in this work, for example:

<https://doi.org/10.1021/nl201874w>

This is an important factor and one we have considered in the past but have not properly mentioned here. The first point is that only the AgNWs were sonicated for different times to achieve size-selection by controlled scission. In contrast, the 2D materials were all sonicated for the same time and size-separated by centrifugation. This means that there is no reason to believe that (for the 2D materials at least) there is any difference in defect density due to variations in sonication time.

In addition, there is much evidence that sonication during liquid phase exfoliation of 2D materials only introduces minimal numbers of basal plane defects. For example, Raman measurements on size-selected liquid phase exfoliated graphene have shown the ratio of the D:G bands to scale perfectly linearly with inverse nanosheet size ($1/L_{NS}$) with an intercept that is very similar to the D:G ratio for the starting graphite (10.1038/NMAT3944). This implies that the defects seen in Raman are predominately edge defects and those basal plane defects that do exist are almost all from the starting graphite (i.e. not created by sonication). Similarly, more detailed Raman studies (10.1016/j.carbon.2024.118801) on similar samples, comparing D:G ratios with G-linewidths concludes that LPE nanosheets have relatively low basal plane defect density. It is also worth mentioning that sonication-exfoliated WS_2 nanosheets show very narrow photoluminescence (PL) linewidths (~ 30 meV) (10.1021/acs.chemmater.9b02905), comparable to those found in mechanically exfoliated samples. As PL spectra tend to broaden in the presence of defects, this is a strong indicator that sonication is not adding large quantities of defects. Finally, PL measurements on MoS_2 nanosheets, exfoliated and size-separated in a manner very similar to the graphene, WS_2 and WSe_2 used here, show that across several size-selected fractions, the PL intensity scales linearly with the population of monolayers in each fraction, indicating that there is no

intrinsic difference between the nanosheets of different sizes (i.e. there is no observable difference in defect content among the different nanosheet sizes).

We have added a version of this argument to “Supplementary note 3. Fitting DC resistivity data” in the SI. We have also added the following text to the main manuscript:

“We argue in Supplementary note 3 that these materials show no significant variations of intrinsic nanosheet properties with size (e.g. due to the presence of sonication-induced defects, that might contribute to the observed size-dependent effects.”

2. In the first paragraph on Page 8, the authors make an assumption: ‘the network carrier density is the same as the nanosheet carrier density’. Even though mobility data (linked to carrier density by equation 1 in the manuscript) presented in the paper may suggest this is true, is this assumption valid, especially for measurements of photoexcited electrons when ‘all networks were thick enough to be in the bulk-like regime’ as stated early in the manuscript? Are there prior works supporting this assertion? At first glance, it seems the literature presents examples of differences semiconducting networks <https://doi.org/10.1021/jp407765r>

The first thing to note is that there are no photoexcited experiments in our current manuscript, so we don’t believe this is a significant issue.

The paper the reviewer refers to relates to charge distributions throughout a 95 μm thick TiO_2 powder network under bright and dark conditions, finding in particular that there is a seven order of magnitude difference between electron transport processes in the powder compared to a single crystal of TiO_2 . The networks of MoS_2 in our work are 5-15 nm thick and, due to the large interfacial area at the junctions, will have a much lower inter-particle resistance compared to a TiO_2 powder. While some charge redistribution may occur within our networks from ambient light, we believe the ultra-thin thickness of our networks, combined with a low junction resistance, mean there should not be any space-charge effects within the network once the device is biased.

Regarding the relationship between network carrier density and nanosheet carrier density, we may not been clear enough. In Supplementary note 1, section 1.2 we argue that the network carrier density is just the nanosheet carrier density rescaled to reflect the

presence of pores: $n_{Net} = n_{NP}(1 - P_{Net})$. In addition, we have now added an alternative derivation to copper-fasten this where we achieve essentially the same result by considering explicitly multiple parallel conductive paths. We have added (see Supplementary note 1, section 1.2 for detailed derivation and discussion).” to the main text to lead the reader to this.

Later, when discussing electrochemically exfoliated nanosheets, we make the assumption that the average carrier density of the nanosheet network is equal to that of an individual nanosheet because the porosity of the network is very low. Previous work on networks of nanocrystals has found that, assuming that all components of a network contribute to the charge transport, the average number of carriers per nanocrystal will equal the average number of carriers across the film (10.1039/C8NR00250A).

To clarify, we have changed the manuscript text and added a new reference to read “the average number of carriers per volume of network is the same as the average number of carriers per volume of nanosheet ([10.1039/C8NR00250A)”

Reviewer #2 (Remarks to the Author):

This is a very thorough investigation on quantifying junction resistance in conductive (semi-conductive) nano-networks.

The authors provide interesting methods to measure the junction resistance in these materials by way of fitting theoretical derivations of the junction's impact on the global resistivity of the samples. This is performed by teasing out the particle resistance and the junction resistance's contribution to overall resistivity depending on temperature and frequency in an EIS setup.

Overall, the contribution of the authors is significant, and very thorough characterizations are provided in the Supplementary Information which is commendable.

Here are my comments and suggestions:

1. On page 3 of the manuscript, the authors mention that previous works exist linking junction resistance to network conductivity but these are too complex for routine utilization. I would like to point out two important things: first, reference 29 utilizes a single equation that only involves physical quantities only and no fitting parameters (the number of junction per wire is known from follow-up work). Second, the present work presents an approximation, which has to be fitted to data first to elucidate fit parameters, therefore cannot be predictive, but only retrospective. Although the derivations are useful, they also rely on a single equation, which does not seem less complex than the mentioned work. Rather than making such sweeping statements about previous work, I would rather highlight this present work in its effort to characterize size dependence of the resistance and novel approaches to tease out R_j v R_{np} .

We apologise, in hindsight our wording was very poor. When we wrote “are too complex for routine utilization” we had in mind equations other than that in ref 29 (e.g. ref 30, 10.1039/C6CP05187A) and our need for data fitting and the conversion of network impedance to nanosheet-junction pair impedance. In addition, we agree that without parameterisation, our model cannot be used for prediction. Prediction was not our intention – we always intended to apply the equations to fitting and impedance analysis. We have tried to improve the wording to fix these problems:

“Another approach to finding R_j involves using models to link network conductivity to junction resistance, generally for nanowire networks.^{29, 30, 31} In particular, these models have allowed the prediction of the electrical properties of some networks²⁹. However, we believe it would be useful to develop simple analytical equations which can be used to fit data for the resistivity of both 1D and 2D networks versus properties such as nanoparticle size, yielding values for R_j and R_{NP} as fit parameters. In addition, access to suitable equations would allow one to directly link the network properties to those of a single (average) nanoparticle-junction pair. As we will show, such a link allows the development of new methodologies to analyse junctions.”

2. I am not convinced by the derivation of σ_{net} (Eq S1). It relies on the assumption that the conductive path is effectively going straight through the sample jumping on a particle, a junction, a particle, a junction, etc. However, previous works show clearly that the conduction paths are influenced by R_j/R_{np} and that depending on these ratios, and the presence of multiple junctions per particle, the current will explore as many junctions as possible (parallel) to minimize dissipated power (i^2r).

We think we have been unclear in our explanation. We believe Reviewer 2 (and Reviewer 3) understand from the text that we are considering a single current path (sometimes referred as a “winner-take-all” path). Having looked back at the text, we can appreciate this misunderstanding – our explanation was lacking. Actually, we imagine the current flowing through a large number of parallel current paths. In fact, if the distributions of nanosheet and junction resistances were narrow enough, these current paths would include all nanosheets. (The impedance data implies such a narrow distribution).

To try to clarify this, we have changed the text in various places where the model is described. For example, we have modified the main text:

“We consider the network as consisting of many well-defined conductive paths in parallel. Within a given current path (Fig. 1a) we assume each carrier passes through a linear array of nanoparticles (Fig. 1b), during which it must cross an inter-particle junction every time it traverses a nanoparticle.”

In addition, we have included a modified derivation in the SI (Supplementary note 1, section 1.2, Method 2) that explicitly considers a large number of parallel paths.

I would urge the authors to set up a simulation of overlapping sheets or wires and check whether S_1 scales correctly on a simulated sample.

Unfortunately, we are not equipped to perform a simulation of overlapping sheets or wires to check whether Equation S_1 scales correctly on a simulated sample at present. However, this might possibly be performed as future work via a collaboration.

Nevertheless, we have performed another check. Ref 29 (10.1021/acsnano.8b05406) develops an equation for the sheet resistance of a nanowire network. As shown in Supplementary note 1, Section 1.7, one can convert this equation into one for the resistivity of the network which can then be simplified by a reasonable approximation. This yields an equation that is very similar to our equation for the resistivity of conducting nanowire networks. This shows that our approach and the approach taken in ref 29 are quite consistent.

We have added the following text to the main manuscript: “While no physics-based models for nanosheet network resistivity exist, we can compare equation 2a to a previously reported model for metallic nanowire networks²⁹. In Supplementary note 1 (section 1.7), we show that the equation for network sheet resistance given in ref²⁹ can be rearranged to give an equation for ρ_{Net} which has properties virtually identical to equation 2a. This supports the validity of our novel approach.”

3. I would be interested in a sensitivity analysis of the estimation of R_j depending on the fitted porosity (results of Fig. 3). How much does R_j depend on the fitting parameters ?

We have performed this analysis and the results have been added to the SI (Supplementary note 3, Fig S16). This analysis shows n_{NS} to be extremely insensitive to the value of P_{Net} used in fitting. R_j was also relatively insensitive, showing a linear variation of $\pm 40\%$ as P_{Net} was varied over a wide range from 0.3 to 0.7.

4. This is minor, but I would not call the EIS approach a direct measurement of R_j since it relies on fitting a Randles circuit to the R_j - R_{np} pair, bringing in two assumptions: that it's a Randles circuit, and C_j as a fit parameter. The idea is still however very interesting.

We have changed the word “direct” to “simultaneous” in the main text.

Overall I would try and structure the writing a little more clearly. Sentences such as

"Measuring R_{NP} and R_J as described above is time-consuming" on page 6 are unclear, and do not naturally bring the author to understand why a new method is presented to measure R_J .

To try to clarify this point, we have changed the text to:

"Measuring R_{NP} and R_J as described above is time-consuming because it requires extensive sample preparation in the form of a size-selection procedure."

In addition, we have gone through the entire text again and made some clarifications.

Reviewer #3 (Remarks to the Author):

This work models the resistance of printed nanomaterial films, finding the nanoparticle and the nanoparticle-to-nanoparticle junction resistance through two different electrical techniques: a D.C. resistivity measurement and an A. C. impedance spectrum. The models used in this work are significant to the printing community to better understand how nanoparticles interact to produce electrically conductive and semiconducting films. This work should be published in Nature Communications after completing the following revisions:

1. How are multiple pathways taken into account in these models? Are you assuming that all of the current is flowing through one pathway? If so, how could this model be expanded to deal with multiple current pathways?

The answer is the same as to a near-identical question from Reviewer 2:

We think we have been unclear in our explanation. We believe Reviewer 3 (and Reviewer 2) understand from the text that we are considering a single current path (sometimes referred as a “winner-take-all” path). Having looked back at the text, we can appreciate this misunderstanding – our explanation was lacking. Actually, we imagine the current flowing through a large number of parallel current paths. In fact, if the distributions of nanosheet and junction resistances were narrow enough, these current paths would include all nanosheets. (The impedance data implies such a narrow distribution).

To try to clarify this, we have changed the text in various places where the model is described. For example, we have modified the main text:

“We consider the network as consisting of many well-defined conductive paths in parallel. Within a given current path (Fig. 1a) we assume each carrier passes through a linear array of nanoparticles (Fig. 1b), during which it must cross an inter-particle junction every time it traverses a nanoparticle.”

In addition, we have included a modified derivation in the SI (Supplementary note 1, p19-20, Method 2) that explicitly considers a large number of parallel paths.

2. What is the distance between the contacts for each material used and does this distance play a factor in the resistance extractions?

In short, our channel lengths are typically between 50 and 200 μm and we have checked does channel length effect the obtained R_j and R_{ns} values and the answer is no.

In more detail, we assume the reviewer is referring to the impedance analysis. The procedure is as follows:

- The measured network impedance is converted to complex network resistivity. This involves using the channel length, L_{ch} (and the film cross-sectional area, A_{Net}).
- Equation 3 (main text) is then used to convert the complex network resistivity to the impedance of a nanosheet junction pair (Z_{NS-J}).

One would expect the impedance of the nanosheet-junction pair, and so the resistance of both nanosheet and junction, to be independent of channel length. We tested this by performing impedance measurements using electrodes with different channel lengths between 50 and 200 μm . As shown in the SI (Supplementary note 11, Figure S24c), both R_{NS} and R_J are independent of channel length. We have added the following text to the main manuscript: “As shown in Supplementary note 11, we found Z_{NS-J} to be independent of channel length (which also allows us to rule out the effects of contact resistance).”

The dimensions of the electrodes used for the DC, AC and transistor measurements in the main text have been added to the “Electrical Characterisation (DC)”, “Electrical Characterisation (AC)” and the “Transistor measurements on a nanosheet network” sections of the Materials and Methods in the SI.

A table of all of constants used in the D.C. and A.C. model such as the area and thickness measurements as well as the extracted values (such as table S1) should be added to the main text as this information is pertinent to the model.

We have added “Table 1” to the main text in line with your suggestion. This table includes the fitting parameters for both the DC network resistivity and AC impedance models used in this work. Fixed/measured values are given in bold in the table, while all extracted values are given with error.

3. Further discussion comparing and contrasting the different materials and their extracted resistances is needed.

We have added the following text to the main manuscript: “This data clearly shows that R_J/R_{NP} values were all >1 indicating that all of these networks were predominately junction-limited. In addition, we can summarise our results for the various materials by plotting R_J versus nanoparticle resistivity, ρ_{NP} , in Fig. 3f. Interestingly, this graph shows a clear relationship between R_J and ρ_{NP} , especially for the 2D materials. The metallic nanoparticles have very low junction resistances of $\sim\Omega$ with the semimetal (graphene) showing $R_J\sim k\Omega$ and the semiconductors $R_J\sim G\Omega$. This implies a relationship between R_J and nanoparticle band structure probably via the details of the inter-particle potential barrier.”

4. Could the A. C. impedance spectrum be applied to conductive material? Is there a reason that this method is specifically used with semiconducting materials?

In principle, impedance spectroscopy can be used to measure the junction resistance for conducting networks. The methodology would be the same but the problem is a technical one. In Figure 3f of the main paper, we show that the junction resistance for conducting networks is considerably lower than that of semiconducting networks. This has a significant impact on the impedance spectrum. All other things being equal, reducing the junction resistance proportionately increases the AC frequency where significant curvature is seen in the $\text{Re}Z$ spectrum. Such curvature is required to achieve fitting to the equivalent circuit equations. Unfortunately for conducting networks, the frequency where this curvature occurs is outside of the range accessible by our instrumentation. However, that does not mean it cannot be measured by other, more sophisticated techniques, e.g. microwave impedance spectroscopy.

Would the temperature dependence measured for the MoS₂ sheets produce similar trends for a conductive material and do the models used for MoS₂ apply for these conductive materials?

Not necessarily. The temperature dependence observed for the MoS₂ nanosheet resistance was consistent with a material with temperature independent carrier density and mobility limited by phonon scattering, specifically acoustic phonons. Where the mobility is limited by impurities or LO phonons, the temperature dependence would be different (see for example [10.1038/s41567-022-01541-y](https://doi.org/10.1038/s41567-022-01541-y)).

The temperature dependence observed for the MoS₂ junction resistance was consistent with hopping which implies $dR/dT < 0$. We would expect similar behaviour for any junctions which consist of a van der Waals gap. Thus, a graphene network should give similar behaviour.

However, if one could measure the temperature dependence of the junction resistance for networks of silver nanowires or silver nanoplatelets, one would expect a different result. For those materials the junctions are thought to consist of fine metallic filaments. Then, the temperature dependence of the junction resistance will have metallic characteristics i.e. $dR/dT > 0$.

5. If both the D. C. and A. C. measurements were taken on the same sample, would the junction and nanoparticle resistances fall within the same value?

We are confident that the answer is yes. We are pursuing this question as future work. However, there are technical difficulties associated with it. The relatively low conductivity of films of liquid-phase exfoliated nanosheets such as those in Figure 3 make the impedance measurement difficult (at least with the equipment we have access to at present). Conversely, for the electrochemically exfoliated nanosheets that we use for impedance spectroscopy, it is quite challenging to produce a set of samples with well-defined and varying nanosheet thickness, as is required to do the DC analysis. This makes it difficult to do both DC and AC measurements on the same material. However, these are technical difficulties that we aim to resolve in the near future.

6. It is stated that the conductivity can be tuned by adjusting material dimensions (in the conclusion), which is not analysed in depth in this work as there is no comparison between different nanoparticle dimensions to show how well this model fits for each dimension parameter considered. Please revise the statement or complete further work.

We have rewritten the conclusion completely, removing this sentence.

7. The supplementary information has a significant amount of information, highlighting how thorough the authors were with this work. However, the main text does not accurately reference or reference enough the findings found in the SI. Please go through and update SI references to be as exact as possible so that people reading this work can easily find important information in the SI, including adding a few words for each SI section referenced. Also, there is information in the SI that is not referred to in the main text. Please refer to all of the SI at least once in the text.

We thank the reviewer for pointing this out. We have gone through the main text and added references to all sections of the SI. In addition, we have also rephrased certain parts for clarity.

8. The addition of an appendix or section of the SI that has abbreviations defined would help readers keep track of variables and abbreviated words. Some variables are not defined in the main text (such as n_{NP}), requiring the reader to search through the SI to find its meaning which can be tedious.

A table containing definitions for all abbreviations used in the main text has been added to the Supplementary Information (Table of Abbreviations, Pages 3-4). In addition, we have added any missing abbreviation definitions, including n_{NP} , to the main text.

9. Change the units of figures 5 c - d to be in units of kelvin rather than Celsius since the other data in this figure is in kelvin.

The units in Figure 5a,c & d have been changed from Celsius to Kelvin.

REVIEWERS' COMMENTS

Reviewer #1 (Remarks to the Author):

The authors have clearly addressed all concerns raised in the initial review I provided. The revised manuscript is an interesting and impactful contribution. Congratulations to the authors.

Reviewer #2 (Remarks to the Author):

I am satisfied with the revisions.

Reviewer #3 (Remarks to the Author):

All concerns have been addressed.